# Oscillatory cortical forces promote three dimensional cell intercalations that shape the murine mandibular arch

Hirotaka Tao[1], Min Zhu[1,2], Kimberly Lau[1], Owen K.W. Whitley[1], Mohammad Samani[1], Xiao Xiao[1], Xiao Xiao Chen[1,3], Noah A. Hahn[1,3], Weifan Liu[1,3], Megan Valencia[3], Min Wu[3], Xian Wang [2], Kelli D. Fenelon[1,3], Clarissa C. Pasiliao[1,3], Di Hu[1], Jinchun Wu[1], Shoshana Spring[4], James Ferguson[5], Edith P. Karuna[6], R. Mark Henkelman[4], Alexander Dunn[7], Huaxiong Huang[8], Hsin-Yi Henry Ho[6], Radhika Atit[5], Sidhartha Goyal[9], Yu Sun[2] & Sevan Hopyan[1,3,10]

Multiple vertebrate embryonic structures such as organ primordia are composed of confluent cells. Although mechanisms that shape tissue sheets are increasingly understood, those which shape a volume of cells remain obscure. Here we show that 3D mesenchymal cell intercalations are essential to shape the mandibular arch of the mouse embryo. Using a genetically encoded vinculin tension sensor that we knock-in to the mouse genome, we show that cortical force oscillations promote these intercalations. Genetic loss- and gain-of-function approaches show that *Wnt5a* functions as a spatial cue to coordinate cell polarity and cytoskeletal oscillation. These processes diminish tissue rigidity and help cells to overcome the energy barrier to intercalation. YAP/TAZ and PIEZO1 serve as downstream effectors of *Wnt5a*-mediated actomyosin polarity and cytosolic calcium transients that orient and drive mesenchymal cell intercalations. These findings advance our understanding of how developmental pathways regulate biophysical properties and forces to shape a solid organ primordium.

[1] Program in Developmental and Stem Cell Biology, Research Institute, The Hospital for Sick Children, Toronto, ON M5G 0A4, Canada. [2] Department of Mechanical and Industrial Engineering and Institute of Biomaterials and Biomedical Engineering, University of Toronto, Toronto, ON M5S 3G8, Canada. [3] Department of Molecular Genetics, University of Toronto, Toronto, ON M5S 1A8, Canada. [4] Mouse Imaging Centre, Hospital for Sick Children, Department of Medical Biophysics, University of Toronto, Toronto, ON M5T 3H7, Canada. [5] Department of Biology, Case Western Reserve University, Cleveland, OH 44106, USA. [6] Department of Cell Biology and Human Anatomy, University of California, Davis School of Medicine, Davis, CA 95616, USA. [7] Department of Chemical Engineering, Stanford University, Stanford, CA 94305, USA. [8] Department of Mathematics and Statistics, York University, Toronto, ON M3P 1P3, Canada. [9] Department of Physics, University of Toronto, Toronto, ON M5S 1A7, Canada. [10] Division of Orthopaedic Surgery, Hospital for Sick Children and University of Toronto, M5G 1X8 Toronto, ON, Canada. Correspondence and requests for materials should be addressed to Y.S. (email: sun@mie.utoronto.ca) or to S.H. (email: sevan.hopyan@sickkids.ca)

Morphogenesis refers to the process of shaping tissue during development, the reproducible nature of which is essential for appropriate pattern formation and function. Most of the recognised principles of morphogenesis concern mechanisms that shape sheets of embryonic tissue[1–6]. In particular, exchange of cell neighbours is central to tissue shape change in two dimensions (2D) and involves a limited number of transient multicellular configurations including tetrads (T1 exchange) and rosettes. Actomyosin contractions generate forces that promote cell neighbour exchanges and are oriented, in part, by physical properties of tissue that may be anisotropic in nature[7–9]. Some of these principles have been extended to curved epithelial sheets by combining empirical and theoretical approaches[10]. However, it remains unclear whether similar mechanisms apply to mesenchymal tissues, in part because mesenchymal tissues have been regarded as less confluent due to the presence of abundant extracellular matrix in some contexts and to potentially less adherent cell–cell junctions[11]. Changes in the viscoelastic properties of tissue are also associated with, and may partly drive, morphogenetic movements[12–15], although the relationship between cellular and tissue scale properties remains incompletely understood and may be context-dependent.

Multiple organ primordia such as the branchial arches and limb buds are composed of an internal bulk layer of mesenchyme. In models of multilayered vertebrate tissues such as the frog gastrula, mechanisms of morphogenesis include amoeboid endodermal cell movements[16] and mesodermal cell intercalations through junctional remodelling, though the latter takes place in a sheet-like manner[17,18]. Another example is elongation of the rod-like skeletal anlage in the vertebrate limb that is attributable to a highly structured columnar arrangement of chondrocytes embedded within abundant extracellular matrix and to oriented rearrangement of nascent daughter cells[19,20]. In contrast to these examples, very little is known about how more-or-less isotropic volumes of confluent cell organise to generate morphogenetic movements. For example, although the directional nature of mesodermal cell movements from the lateral plate to the limb bud has been demonstrated[21–23], the cellular basis of those movements, and of volumetric morphogenesis in general, remain largely uncharacterised.

Early branchial arches are composed of a core volume of mesenchyme that is surrounded by a single cell layer epithelium. Although the neural crest[24] and cranial mesodermal[25] origins of branchial arch mesenchyme are well recognised, mechanisms by which the structure grows outward and acquires shape are less clear. Loss-of-function approaches that were intended to define the roles of various signalling pathways identified cellular processes that are relevant to branchial arch morphogenesis, such as cell survival[26,27], cell proliferation[28], and cell migration[29]. Those studies generated important insights, but were not intended to explain how the branchial arches acquire shape.

Craniofacial anomalies are common birth defects[30]. The mandibular portion of the first branchial arch generates multiple facial structures including the lower jaw, and some syndromes are associated with structurally common features of malformation such as a short (front to back) and broad (side to side) mandible. One of these, Robinow syndrome, is caused by mutations in components of the noncanonical WNT pathway including WNT5A (autosomal-dominant form) and ROR2 (recessive form) which encode a ligand and a downstream receptor tyrosine kinase, respectively[31–33]. Wnt5a[32,34] and Ror2[35,36] mutant mice that phenocopy many of the features of the human syndrome exhibit a short and broad mandibular arch. Facial anomalies also result from mutations of Dachsous1 and FAT4 in autosomal recessive Van Maldergem and Hennekam syndromes[37,38]. These genes encode a receptor-ligand cadherin pair that regulates planar

cell polarity (PCP) and are upstream of yes-associated protein (YAP), a transcriptional effector of the Hippo pathway[39]. Autosomal recessive mutations of piezo type mechanosensitive ion channel component 1 (PIEZO1) that encodes a pore forming subunit of a mechanically activated ion channel also cause facial dysmorphism with hypoplastic features that overlap with those above[40,41]. The cellular and biophysical basis of these malformations are not clear, although it has been shown that mutant WNT5A may exhibit neomorphic properties that affect cell polarity and migration in a chick model of human Robinow syndrome[42].

Here we study the mandibular arch as a model of two distinct modes of 3D morphogenesis. We show that cell division and tissue-scale physical properties are important for growth but do not sufficiently explain how the arch primordium acquires a narrow mid-portion and a bulbous distal portion. Our data support a model in which 3D mesenchymal cell intercalations narrow and elongate the mid-portion. Relatively high amplitude cortical force oscillations and cell polarity promote cell intercalations in a Wnt5a-dependent, Yap/Taz-dependent, and Piezo1-dependent manner, implying these regulators spatially fine tune physical cell behaviours.

## Results

**Cell division does not explain mandibular arch shape.** The first branchial arch buds at the ventral aspect of the midbrain-hindbrain boundary in the mouse embryo. As shown by optical projection tomography (OPT), the mandibular portion of the first branchial arch remodels from a rod-like structure at the 19 somite stage (~E9.0) to form a narrow central waist and a distal bulbous region between somite stages 21–28 (~E9.25–E9.75; Fig. 1a). In this study, we focused on the shape change between 19 and 21 somite stages (a ~4 h period) (Supplementary Movie 1, 2) to understand how the waist and bulbous regions become defined.

To examine whether spatial variation in the frequency of cell division influences tissue shape, we measured cell cycle times using dual pulse with two thymidine analogues (5-chloro-2′-deoxyuridine/5-iodo-2′deoxyuridine (CIdU/IdU))[43]. Each of the epithelial and mesenchymal tissue layers were arbitrarily divided into twelve spatial regions within which mean cell cycle times were calculated (Fig. 1b, Supplementary Fig. 1A). Cell division was most rapid in the proximal third of the mandibular arch adjacent to the face but there was little difference between the middle and distal thirds (Fig. 1c, d), implying this parameter does not intuitively explain differences in waist/bulbous morphology.

Physical tissue properties profoundly influence morphogenesis. Using atomic force microscopy (AFM) to measure elasticity (Fig. 1e, Supplementary Fig. 1B), we performed shallow (1 μm) indentation of the ~12 μm thick epithelium to exclude substrate effects of the underlying mesenchyme (Supplementary Fig. 1C) and found that mid-portion epithelium becomes stiffer relative to more proximal and distal flanking regions between somite stages 19 and 21 (Fig. 1f). Mesenchymal stiffness was measured using deep (7–9 μm) indentation (Supplementary Fig. 1D) and decoupled from the influence of the epithelium (see Methods section). In contrast to the epithelium, mesenchyme was least stiff in the middle region (Fig. 1f). Under shallow indentation, the arch behaved as an elastic material (Supplementary Fig. 1C), whereas deep probe indentation and retraction revealed evidence of hysteresis, or dissipation of energy, an indication of viscoelasticity (Supplementary Fig. 1D). Combined (epithelial and mesenchymal) tissue viscosity that was quantified using different rates (5, 10, and 15 μm/s) of indentation (Supplementary Fig. 1E) also increased over developmental time but did not vary spatially throughout the arch (Fig. 1g). An unbiased model would

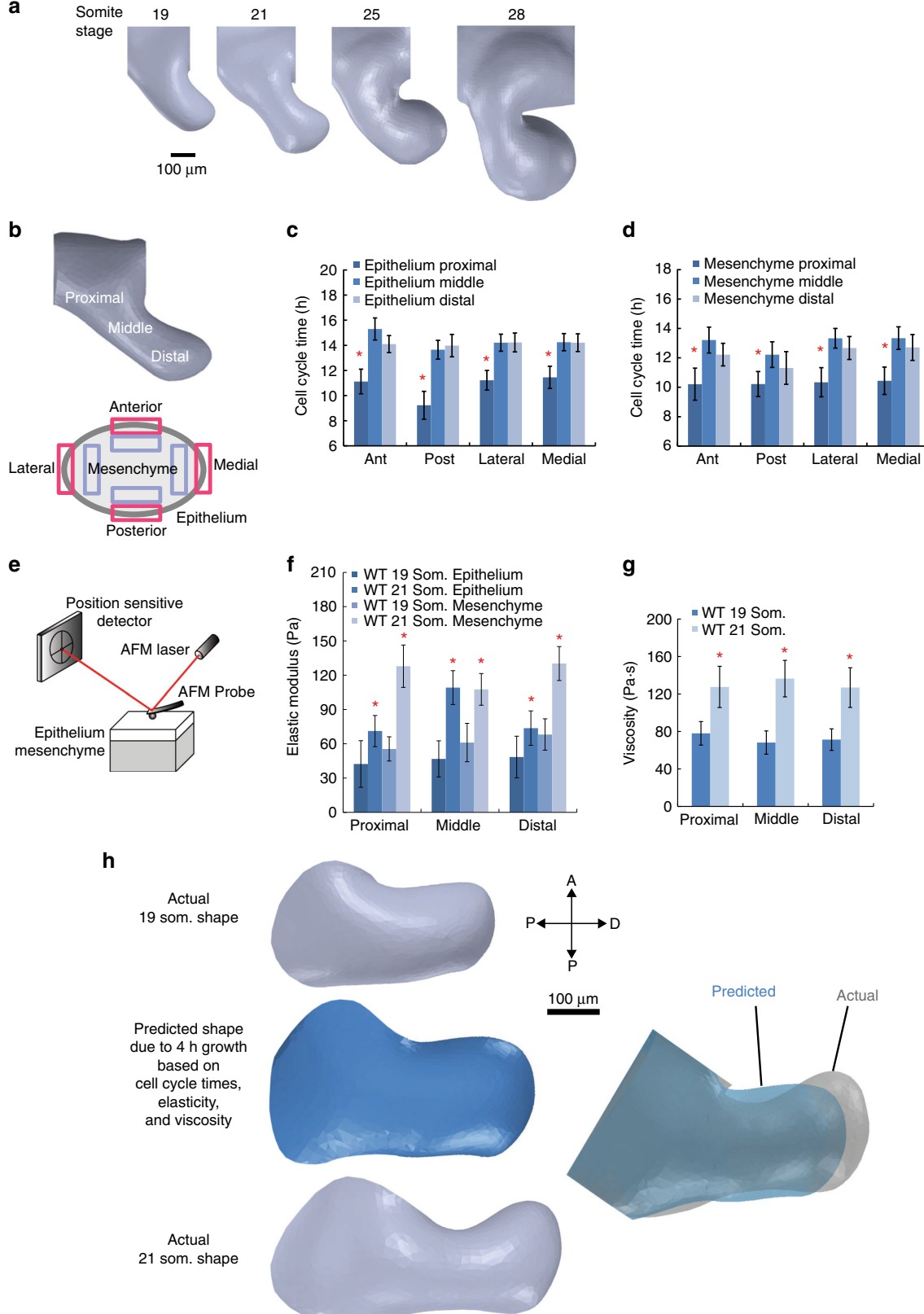

be useful to evaluate the potential morphogenetic relevance of these data.

To integrate the influence of cell divisions and physical properties on tissue shape, we constructed a 3D, two tissue layer finite element model of the 19 somite stage mandibular arch that was based on OPT-derived tissue topography (Supplementary Fig. 1F, G), an approach conceptually similar to that previously reported for limb bud shape[44] but with the addition of empirical physical properties. Based on prior work, we initially hypothesised that disproportionately increased stiffness of the middle epithelium might resist hoop stress due to growth of the mesenchyme to yield a growth process reminiscent of directed

**Fig. 1** Cell cycle times and tissue properties are insufficient to explain mandibular arch shape. **a** Sagittal OPT renderings of the right mandibular arch in the mouse embryo at different stages. **b** Cell cycle time was measured in 24 adjacent regions (4 epithelial and 4 mesenchymal regions each within proximal, middle and distal arch) of the 19 somite stage mandibular arch. **c**, **d** Spatial variation of epithelial (**c**) and mesenchymal (**d**) cell cycle times in the mandibular arch. Cell division was more rapid in the proximal region for both epithelial and mesenchymal layers; $n = 3$ embryos at 20 somite stage, 15–35 cells examined for each of 12 epithelial regions per embryo, 50–75 cells examined for each of 12 mesenchymal regions per embryo; asterisks denote $p <$ 0.05, Student's $t$-test, error bars denote standard error of the mean (s.e.m.). **e** Tissue indentation by AFM was employed to measure properties of intact, live mouse embryos. **f** Elastic (Young's) modulus (stiffness) of epithelium and mesenchyme. **g** Viscosity of whole tissue in proximal, middle and distal regions of the mandibular arch at 19 and 21 somite stages. For **f** and **g**, 15 separate sites in each proximal, middle and distal region were indented in triplicate (45 measurements per region) per embryo; $n = 2$ embryos per condition, asterisks denote statistical significance, p value range: $10^{-6}$ to $10^{-19}$, two-tailed $t$-test, error bars denote standard deviation. **h** Finite element simulation of 4 h of growth beginning from the actual 19 somite stage mandibular arch shape to predict 21 somite stage shape. The model incorporated experimentally measured spatial variation of cell cycle time, elasticity and viscosity. Simulated growth (in blue) results in an arch that is shorter and broader than the actual 21 somite stage arch. Source data are provided as a Source Data file

dilation[45,46]. However, simulated growth and deformation based on the spatial variations of cell cycle time, elasticity and viscosity predicted an inappropriately short and wide arch (Fig. 1h, Supplementary Movie 3). We reasoned that the inadequacy of this continuum model likely reflects the fact that it cannot account for defined cell rearrangements.

**Distinct growth patterns characterise two regions of the arch.** In confluent 2D epithelia, defined neighbour exchange processes are essential for tissue remodelling. We noticed that the mesenchymal cells within the mandibular arch exhibited key characteristics of mouse epithelial cells that rearrange through neighbour exchange, such as confluence, abundant expression of cell–cell junction proteins (N-cadherin and desmoglein), and protrusive activity (Supplementary Fig. 2A–C)[7,47].

To examine dynamic cell behaviours in 3D, we performed lightsheet microscopy of live, intact mouse embryos that harboured CAG::H2B-GFP and mTmG transgenic reporters to highlight nuclei in green and cell membranes in red (Supplementary Movie 4, 5). Since this method of imaging the organ stage mouse embryo has not been previously used, we examined morphogenesis under different conditions. Increasing agarose concentration and the absence of serum in the culture medium did not result in greater apoptosis during 4 h culture (Supplementary Fig. 2D). However, appropriate diminution over time of the width/length ratio of the arch was slowed the most when it was cultured in 2% agarose compared to development in utero (~2 h per somite) (Supplementary Fig. 2E). One percent agarose with media containing serum was chosen for live imaging experiments to minimise and facilitate tracking of tissue drift by embedding fluorescent beads alongside the embryo during 2–3 h time-lapse sessions.

Since biological parameters that are relevant to the possibility of 3D cell intercalation have not been well defined, we explored ideas from the physical sciences that are hypothetically relevant to morphogenesis. As in an unstable foam[48,49], mesenchymal cells exhibited a broad distribution of cell faces (7–14) with relatively few neighbours in the mid-portion of the mandibular arch (Fig. 2a, b, Supplementary Fig. 2F, Supplementary Movie 6), suggesting cells in that region are furthest from equilibrium and most likely to exchange neighbours. To estimate cell shapes, we segmented nuclei in 3D to act as centroids for Voronoi tessellation (Fig. 2c). According to a recent model, there is a relationship between the rigidity of confluent 3D tissues and the ratio of cellular surface area to volume[50]. The cell shape parameter from that model, surface area/volume$^{2/3}$ ($S/V^{2/3}$), varied along the proximodistal axis of the mandibular arch with the highest values observed in the middle region (Fig. 2d, e) that is consistent with a relatively more liquid behaviour compared to proximal and distal regions. These parameters predict that middle region cells should preferentially engage in neighbour exchange.

To identify the large-scale pattern of tissue displacement, a subset of fluorescently labelled nuclei were tracked in 4D after accounting for embryo drift by marking multiple fluorescent beads that we co-embedded with the embryo in an agarose cylinder. Tissue growth was visibly more longitudinally oriented in the middle region compared to the distal region (Fig. 2f, g, Supplementary Movie 7, 8). Using a random walk model, we found that the cumulative distribution function (CDF) of persistence time (the extent to which cells move in accordance with their recent past[51,52]) exhibited a greater slope for cells in the middle compared to those in the distal region, indicating the former are more directionally consistent over time (Supplementary Fig. 2G–J, Supplementary Movie 9, 10). Therefore, distinct morphogenetic movements characterise the middle and distal regions of the arch.

**Cell intercalations promote volumetric convergent extension.** Cell displacements, by themselves, are not evidence of active cell movements since the tissue they inhabit expands due to cell divisions. Analysis of tissue strain indicated that proximodistal elongation of the arch was accompanied by compression along the short rostrocaudal axis of the midportion (Fig. 2h, i, Supplementary Movie 11), an observation that can potentially be explained by convergent cell movements. To test for this possibility, we first examined epithelial cells at higher resolution using live confocal time-lapse imaging. In the middle epithelium, although the alignment of daughter cells was biased along the short rostrocaudal axis, T1 exchange events tended to shorten the rostrocaudal axis and elongate the proximodistal axis of growth. Epithelial cell rearrangements were less oriented in the distal region (Fig. 3a, b, Supplementary Fig. 3A), consistent with bulbous growth.

For mesenchyme, live light sheet movies at whole tissue and intermediate scales (~100 cell volumes) revealed that cells in the middle region converged centripetally relative to one another as tissue growed longitudinally (Fig. 3c, Supplementary Movie 12, 13). At small scales (~10–15 cells), the intercalation of a single cell into a nest of 5–7 others was the smallest multicellular unit in which 3D neighbour exchange was observed (Fig. 3c, Supplementary Movie 14–16). This configuration is analogous to 3D T1 exchange in an unstable foam[49]. Tracking of nuclear centroids undergoing intercalation confirmed that small groups of cells converged in the axial plane as the tissue extended distalward (Supplementary Fig. 3B). In contrast, more subtle intercellular adjustments characterised the distal region (Fig. 3c, Supplementary Fig. 3C, Supplementary Movie 17). These observations support the concept that regional differences in cell intercalation correlate with large-scale growth patterns and that volumetric convergent extension elongates the middle region (Fig. 3d).

Cortical actomyosin can orient forces that drive cell intercalation in epithelia and in mesoderm[17,53,54]. F-actin and phospho

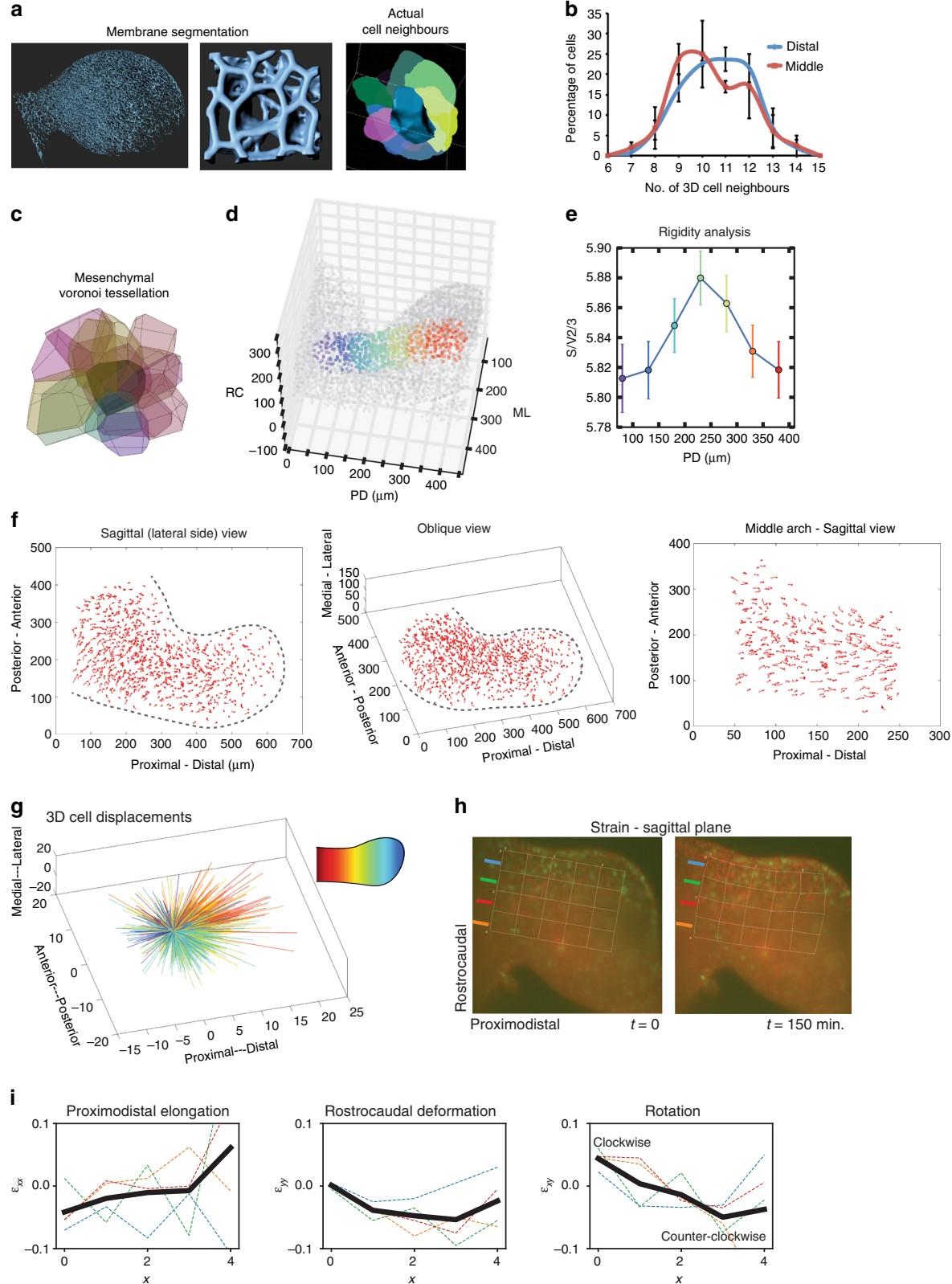

(p)-myosin light chain (pMLC) were biased parallel to the short, transverse axis among epithelial and mesenchymal cells in the middle waist, but not proximal and distal regions, of the arch (Fig. 3e, Supplementary Fig. 3D). This bias likely reflects the tissue stress pattern[7] and is consistent with the axis of intercellular movements observed in the waist region, but itself does not explain why cells intercalate.

**Cortical forces correlate with mesenchymal intercalations**. Cortical tension is a key parameter that regulates cell sorting and intercalation during development[55,56]. Local actomyosin abundance or cell interface length changes have been used to estimate relative cortical tension, but may not reflect actual forces or their dynamic fluctuations. A more direct proxy for cortical tension would be force experienced by the cortical protein vinculin, the

**Fig. 2** Two distinct patterns of growth characterise the mandibular arch. **a** 3D renderings of cell membranes labelled by *mTmG* based on live light sheet microscopy. Whole arch (left) and local cell neighbour relations (middle and right with each colour representing one cell) are shown. Scale bar: 40 μm. **b** Distribution of numbers of cell neighbours in middle (red curve) and distal (blue curves) mandibular arch (n = 2 embryos, 202 middle cells and 244 distal cells examined, p < 0.05, chi-squared; scale bar: 40 μm. **c** Voronoi tesselation of mesenchymal nuclei to estimate cell shapes. **d, e** Colour-coded spatial distribution of cell shape index ($S/V^{2/3}$) that correlates positively with liquid-like tissue phase (greatest in the middle region). **f** 4D tracks of a subset of mandibular arch cells are shown in two orthogonal views. Relatively directional tracks that were oriented distalward characterised the waist region, whereas short and tightly curved tracks characterise the bulbous region (representative of 3 embryos at 20 somite stage). **g** 3D dandelion plot of spatially colour-coded trajectories of cells at the start and end of a movie. Waist cells (red/orange) move predominantly outward whereas bulbous cells move in a relatively radial fashion. **h** Strain illustrated as deformation of a sagittal plane grid during a 150 min. movie. Nodes remain fixed to the same positions throughout the movie. Coloured lines to the left of each grid correspond to rostrocaudal rows of squares that correspond to dashed lines in **i**. Corresponds to Supplementary Movie 11. (Representative of 3 embryos at 20–21 somite stage.) **i** The proximodistal axis elongated (reflected by the positive $\varepsilon_{xx}$ slope), while the midportion of the arch converged along the rostrocaudal axis (reflected by negative values of the $\varepsilon_{yy}$ curve). Convergence of midportion tissue combined with expansion of distal tissue results in clockwise (positive $\varepsilon_{xy}$ values) and counter-clockwise (negative $\varepsilon_{xy}$ values) rotational deformation of adjacent regions, respectively (as reflected by the downward $\varepsilon_{xy}$ slope). Dashed, coloured lines correspond to rostroacaudal rows of grid components of the strain tensor as depicted in **h**. The solid black line is the average of rostrocaudal strain tensors as it varied along the proximodistal axis. Source data are provided as a Source Data file

head and tail domains of which link cadherin/catenin complexes at adherens junctions with cortical actin, respectively[57]. To directly measure this parameter, we knocked-in a conditional Förster resonance energy transfer (FRET)-based vinculin tension sensor (VinTS), for which fluorescence lifetime in nanoseconds (ns) is proportional to force in piconewtons (pN)[58], into the mouse *Rosa* locus. We generated two control knock-in strains that should exhibit maximal (donor only VinTFP—no FRET), and minimal (vinculin tailless VinTL—maximal FRET due to lack of C-terminal actin binding sites) fluorescence lifetime, respectively (Fig. 4a).

All three knock-in constructs were expressed robustly in the appropriate cortical domain in vitro (Supplementary Fig. 4A) and in vivo (Fig. 4b, Supplementary Movie 18). Measurements of the dynamic range, floor and ceiling lifetime values of the three strains were undertaken in ES cell colonies and embryoid bodies prior to their evaluation in the mouse embryo under conditions that we previously optimised for confocal live imaging[7,23]. Single cell cortices were outlined as regions of interest to measure lifetime. As expected, the donor-only VinTFP construct reported the longest lifetime with a narrow standard deviation and VinTL exhibited short lifetime values. In embryoid body cells and among differentiated beating cardiomyocytes, the range and standard deviation of the full length VinTS lifetime values was greater than that of either control suggesting the reporter was responding dynamically to cell contractions (Supplementary Fig. 4B, C). Among epithelial and mesenchymal cells of the mandibular arch, VinTS reported greater dynamic lifetime range and standard deviation than either control strain indicating robust dynamic capacity in vivo (Supplementary Fig. 4D). Treatment of live transgenic embryos with the Rock inhibitor Y27632 or the actin polymerisation inhibitor Cytochalasin D lowered average VinTS lifetime values to floor levels for this construct and dampened standard deviation values measured in the mandibular arch (Supplementary Fig. 4E). These data indicate the VinTS sensor reports forces attributable to actomyosin contraction in vivo.

For both epithelial and mesenchymal tissue layers, cells in the middle region exhibited lower average cytoskeletal tension along with greater variability compared to the corresponding distal region (Fig. 4c). Since high cortical tension generally reflects relatively spherical cell shapes, these differences are consistent with cell face numbers we documented and support the concept that cells in the middle region are more likely to intercalate. Time-lapse analysis with two-minute intervals revealed an oscillatory pattern of fluorescence lifetime fluctuation among individual cell cortices using the VinTS reporter. In support of a previous theoretical assertion that membrane fluctuation is necessary to overcome the energy barrier for cell intercalation[12,13], middle region mesenchymal cells exhibited a significantly greater fluctuation amplitude compared to those in the distal region (Fig. 4d, e).

Transient increases in cytosolic $Ca^{2+}$ ion concentration are required for contractile cortical pulses. We applied a $Ca^{2+}$ indicator (X-rhod-1 or Fluo8) to intact mouse embryos and quantified the proportion of X-rhod-1 fluorescence intensity to total cell area (marked by VinTS) over time. X-rhod-1 intensity fluctuated to a greater extent among middle region mesenchymal cells than in distal cells (Supplementary Movie 19, 20, Supplementary Fig. 4F). Also, changes in X-rhod-1 intensity better correlated with changes in VinTS fluorescence lifetime among cells in the middle region of the arch (Fig. 4f), suggesting $Ca^{2+}$ fluctuation promotes cortical oscillation.

**Wnt5a regulates cortical polarity and oscillation.** We observed that $Wnt5a^{-/-}$ mouse embryos, which substantially phenocopy Robinow syndrome[32,34], exhibit a proximodistally short and mediolaterally broad mandibular arch (Fig. 5a, Supplementary Movie 21, 22). Cell cycle times in $Wnt5a^{-/-}$ embryos were similar to those of WT embryos (Supplementary Fig. 5A, B, compare with Fig. 1c, d), although stiffness and viscosity did not increase to the same degree as in WT tissue, especially in the middle region (Supplementary Fig. 5C, D, compare with Fig. 1f, g). Finite element simulation of mandibular arch growth more closely matched actual mutant arch shape as compared to the WT simulation (Fig. 5b, c, Supplementary Movie 23), suggesting that the influence of cell rearrangements might be less relevant to growth in the $Wnt5a^{-/-}$ mutant. Consistent with that concept, mutant cells exhibited directionally less persistent movements that were reflected by a lower slope for cumulative distribution function (CDF) for both persistence time and angle from the mean compared to cells in WT embryos (Fig. 5d, e).

In $Wnt5a^{-/-}$ mutants, live light sheet imaging revealed that 2D epithelial T1 exchanges were disoriented (Fig. 5f, g, compare with Fig. 3a). Mesenchymal cells exhibited a rightward-shifted (more stable) distribution of cell neighbour numbers (Fig. 5h), suggesting they are less likely to rearrange. Indeed, the movements of $Wnt5a^{-/-}$ middle arch cells were more radially oriented compared to WT cells (Fig. 5i, Supplementary Movie 24, 25, compare with Fig. 2f, g), and the frequency of mesenchymal intercalations was diminished (Supplementary Fig. 5E, Supplementary Movie 26). Cortical F-actin, phosphomyosin light chain (pMLC), and VANGL2 were not biased along rostrocaudal cell interfaces in $Wnt5a^{-/-}$ mutants (Fig. 6a, Supplementary Fig. 6A–C). Moreover, the amplitudes of cortical oscillations (Fig. 6b–d) and $Ca^{2+}$ fluctuations (Fig. 6e, Supplementary

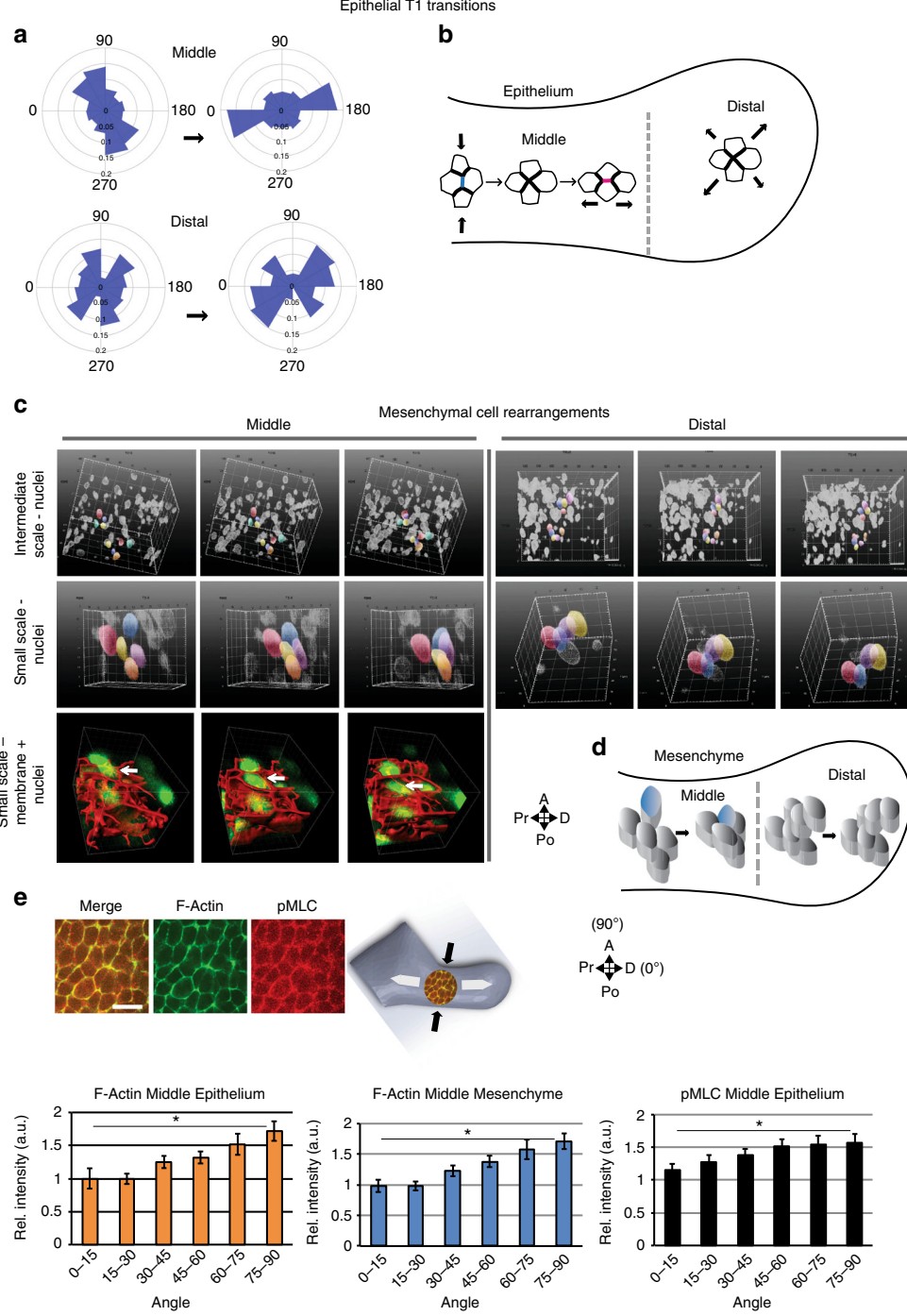

**Fig. 3** Epithelial and mesenchymal cell rearrangements converge and extend the midportion of the mandibular arch. **a** Orientation of epithelial tetrads during T1 transitions at formation to resolution stages (separated by an arrow); $n = 3$ embryos at 21 somite stage. **b** Schematic representation of predominant orientation of epithelial T1 transitions in middle and distal regions. **c** Time series (taken from 120 min. time lapse moves) of volumes of mesenchymal cells of dual *H2B-GFP;mTmG* transgenic embryos visualised by light sheet microscopy at intermediate and high magnification. Select nuclei are coloured to show tissue and cell convergence at intermediate and small scales occurs in the middle, but not distal, region. (Representative of 5 embryos at 19–21 somite stage). **d** Schematic representation of oriented mesenchymal cell intercalations transverse to the axis of elongation in the middle region. **e** In the mid-portion of the arch, F-actin and phosphomyosin light chain (pMLC) were biased along proximal and distal epithelial and mesenchymal cell interfaces which is parallel to the rostrocaudal axis and to the direction of cell intercalations. The angular distribution of immunostain fluorescence intensity for epithelial ($n = 4$ embryos) and mesenchymal ($n = 4$ embryos) F-actin and epithelial phosphomyosin light chain (pMLC) ($n = 5$ embryos) relative to the arch long axis that was designated as 0° was quantified in the middle region using SIESTA. Scale bar: 20 μm, asterisks denote $p < 0.05$, Student's *t*-test, error bars denote s.e.m. Source data are provided as a Source Data file

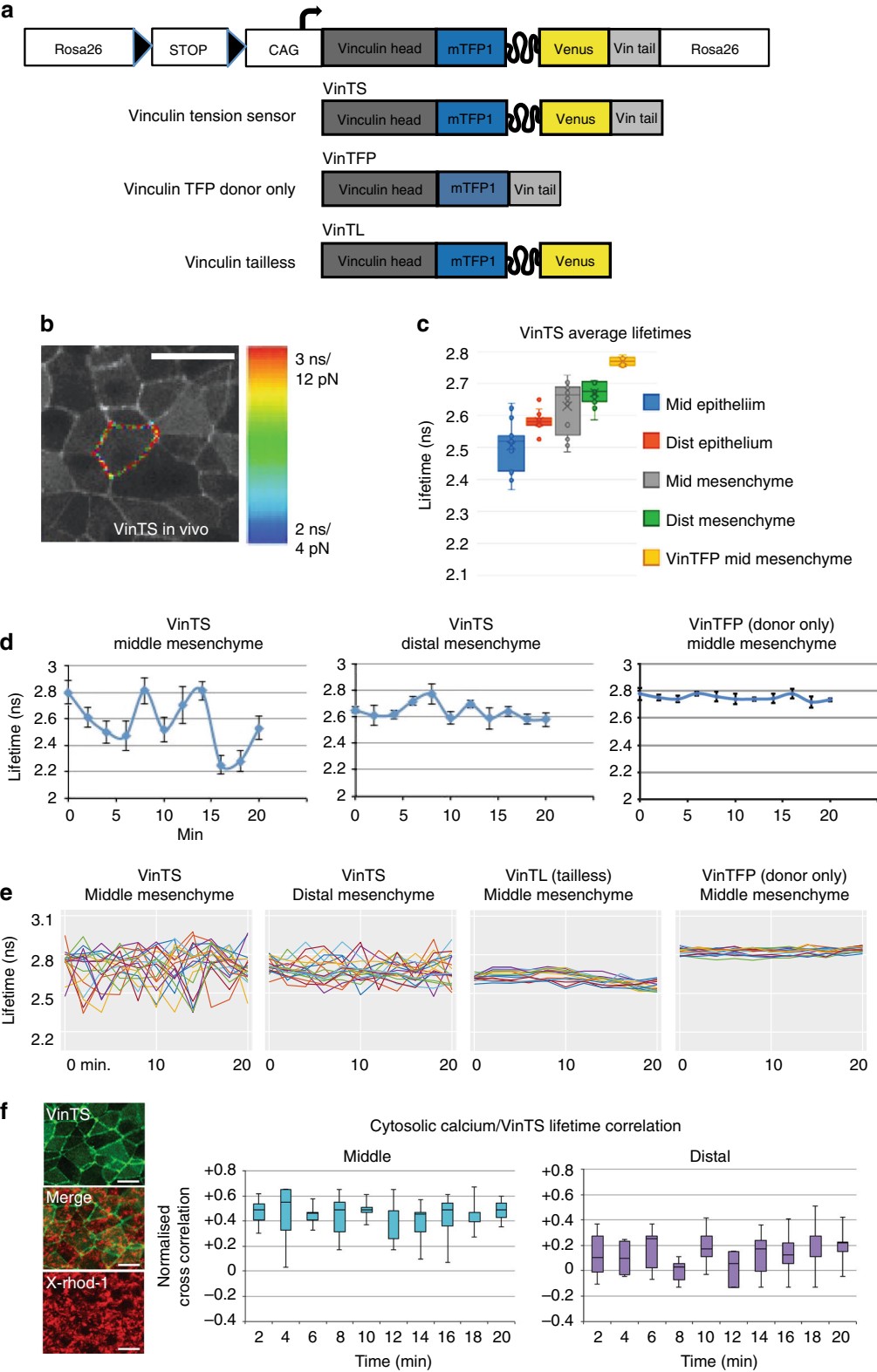

Fig. 6D, Supplementary Movie 27, 28) were diminished in the mutant middle arch. Together, these data imply that *Wnt5a* acts through the cell polarity and $Ca^{2+}$ pathways to promote and orient cell intercalations.

The *Wnt5a* expression domain is biased distally and diminishes steeply within the middle region of the arch, but it was not clear whether it acts instructively or permissively to influence cortical behaviour. To test between these possibilities, we overexpressed a transgenic *Cre*-activated *Wnt5a* allele that was targeted to the endogenous *Ubiquitin-b* (*Ubb*) locus using *Sox2:Cre* to drive ubiquitous expression and diminish the *Wnt5a* gradient in the mandibular arch (Fig. 7a, Supplementary Fig. 7A).

**Fig. 4** Vinculin force oscillations distinguish middle and distal regions of the mandibular arch. **a** Conditional *Rosa26* knock-in mouse strains: full length vinculin tension sensor (VinTS), TFP (FRET donor) only control (VinTFP), vinculin tailless control (VinTL). **b** Tension sensor expression among epithelial cells in the mandibular arch with one cell cortex highlighted as region of interest. Colour scale shows range of lifetime (in nanoseconds, ns) and corresponding force values (in picoNewtons, pN). **c** Individual cell fluorescence lifetime values in middle (mid) and distal (dist) epithelium and mesenchyme of the mandibular arch. Boxplots show mean (x), median (---), central quartiles (coloured box), and range (transverse end bars); $n = 15$ cells per region in each of 3 embryos. **d** Representative vinculin force curves of individual cells in middle and distal regions, and donor only control. Lifetime readings were taken at two minute intervals, error bars denote s.e.m. **e** Multiple vinculin force curves. The sample variance of lifetime values, a measure of amplitude, was greater among middle (0.0201 ns) versus distal (0.0132 ns) mesenchymal cells ($n = 5$ embryos, 15 cells per embryo per condition, $p = 0.03$, t-test). Mean lifetime variance was lower among middle mesenchymal cells of control VinTL (0.0028 ns, $p = 0.02$, ANOVA) and VinTFP (0.0006 ns, $p = 0.01$, $n = 3$ embryos, 15 cells per embryo per condition) strains. **f** Correlations of VinTS fluorescence lifetime and calcium reporter X-rhod-1 fluctuation (defined as the percentage of X-rhod-1-positive area for each cell at each time point) in middle and distal arch regions over time (18 cells per region). Source data are provided as a Source Data file

In *Sox2:Cre;Z/Wnt5a* embryos, the mandibular arch was short and broad (Fig. 7b, Supplementary Movie 29), and lacked actomyosin polarity (Fig. 7c), implying that the native *Wnt5a* expression domain provides a spatial cue for cortical organisation.

**Wnt5a acts upstream of YAP/TAZ and PIEZO1**. We examined the Hippo pathway since it has been implicated in craniofacial malformation[37,38], it is mechanoresponsive[59], and is recognised in vitro as a downstream effector of noncanonical WNT5A signalling that promotes YAP/TAZ nuclear accumulation[60]. In *Wnt5a* loss of function mutants, nuclear localisation of YAP was indeed diminished among mesenchymal cells of the middle and distal region of the branchial arch (Fig. 7d). We used *T:Cre* or *Wnt1:Cre* to excise *Yap/Taz* in early mesodermal somitomeres (~E7.5) or in neural crest, respectively, both of which contribute to branchial arch mesenchyme. Although deletion of all four *Yap/Taz* alleles caused lethality prior to branchial arch development, removal of any three *Yap* and *Taz* alleles with either Cre grossly phenocopied *Wnt5a* null mutants with respect to mandibular shape (Fig. 7e). Loss of the upstream Hippo regulator *Fat4* had no impact on mandibular arch size or shape (Supplementary Fig. 7B), -suggesting the possibility that YAP/TAZ act primarily downstream of WNT5A in this context. Unlike in *Wnt5a* mutants, intercellular movements were present but seemed inconsistent and were disoriented among *Yap/Taz* mutants (Fig. 7f, Supplementary Movie 30). $Ca^{2+}$ fluctuation was diminished among middle arch cells of *Yap/Taz* mutants (Supplementary Fig. 7D), and cortical forces fluctuated inconsistently from fairly flat in some cells to normal amplitudes in others (Fig. 7g). Also, disrupted actomyosin polarity (Supplementary Fig. 7C) likely contributed to disoriented cell rearrangements. Possible explanations for the persistence of cortical oscillations among a portion of mesenchymal cells include the influence of other ions or regulators of oscillation and the influence passive membrane shape changes secondary to disorderly cell rearrangements. These data imply that cortical polarity and cortical oscillation are regulated by *Yap/Taz* in concert with or downstream of *Wnt5a*.

To further test whether cytosolic $Ca^{2+}$ transients contribute to mandibular arch morphogenesis using another genetic perturbation, we examined *Piezo1* which encodes a mechanosensitive ion channel[61]. As expected, $Ca^{2+}$ fluctuation was diminished in $Piezo1^{-/-}$ mutants (Supplementary Fig. 7E, Supplementary Movie 31). PIEZO1 immunostain intensity was diminished in $Wnt5a^{-/-}$ and *Yap/Taz* mutants (Fig. 7h, Supplementary Fig. 7F - antibody specificity was confirmed by its absence in *Piezo1* mutants), indicating its expression or stability is regulated by a *Wnt5a-Yap/Taz* pathway. Interestingly, nuclear abundance of YAP was diminished in the absence of *Piezo1* (Supplementary Fig. 7G), possibly reflecting the importance of cytoskeletal,

mechanical cues for YAP localisation. Similar to *Wnt5a* and *Yap/Taz* mutants, actin bias along rostrocaudal interfaces was lost in $Piezo1^{-/-}$ mutants (Supplementary Fig. 7H). The mandibular arch of $Piezo1^{-/-}$ embryos was misshapen similar to that of $Wnt5a^{-/-}$ mutants (Fig. 7i, Supplementary Movie 32), suggesting cell intercalations are disrupted. Together, these findings implicate YAP/TAZ and PIEZO1 as downstream effectors of WNT5A-mediated cortical polarity and oscillation, and indicate cross-talk between these regulators (Fig. 7j).

## Discussion

Our findings suggest that cell intercalations shape a volume of confluent cells, and that basic modes of 2D cell rearrangement, such as T1 exchange, have 3D counterparts previously observed in foams that remodel mesenchyme. Step-wise evolution of cell intercalation capacity from in-plane among diploblasts[62], to out-of-plane among triploblasts[63], to within a volume of cell neighbours may have facilitated the radiation of increasingly complex body plans among Bilateria and vertebrates.

By acting partly in the same pathway, *Wnt5a*, *Yap/Taz*, and *Piezo1* transform biochemical signals to mechanical regulation of two key cellular parameters. These include changes in geometry that make cells more elongate and less densely packed as a function of cell polarity, and effectively lower the energy barrier to intercalation. Also, the oscillatory nature of cytoskeletal contractions that have been observed in association with multiple types of invertebrate and vertebrate cell movements[6,53,64] helps to push cells over the energy barrier to intercalation[12,13]. Cell rearrangements, as in the middle arch, lower the viscosity of tissue, allowing it to flow in a liquid-like manner[14]. As cell rearrangements cease, tissue gradually stiffens as has also been observed during dorsal closure of the amnioserosa in *Drosophila*[65] and the zebrafish tailbud[14]. A different suite of cell and tissue properties, possibly related to extracellular matrix and tissue differentiation, may regulate the morphogenesis of solid embryonic structures and organs at later stages. During primordial stages, we propose that *Wnt5a*, YAP/TAZ, and PIEZO1 control the liquid-to-solid transition of tissue properties by coordinating cell polarity with cortical oscillations to spatiotemporally orient cell intercalations.

Explanations for the oscillatory nature of contractions that have been put forward include cell-extrinsic[66] and cell-intrinsic mechanisms[64,67–69]. Ion channels drive $Ca^{2+}$ fluctuation and periodic cell contraction in *Drosophila*[70], a function that we find is partially fulfilled by *Piezo1* in mouse mesenchyme. Our study emphasises oscillation amplitude, although oscillation frequency has also been implicated in the regulation of convergent extension[71]. Feedback between positive and negative regulators of $Ca^{2+}$ influx, such as the noncanonical Wnt pathway and $Ca^{2+}$-dependent proteases like calpain-2, respectively, have the potential to regulate oscillation amplitude and frequency.

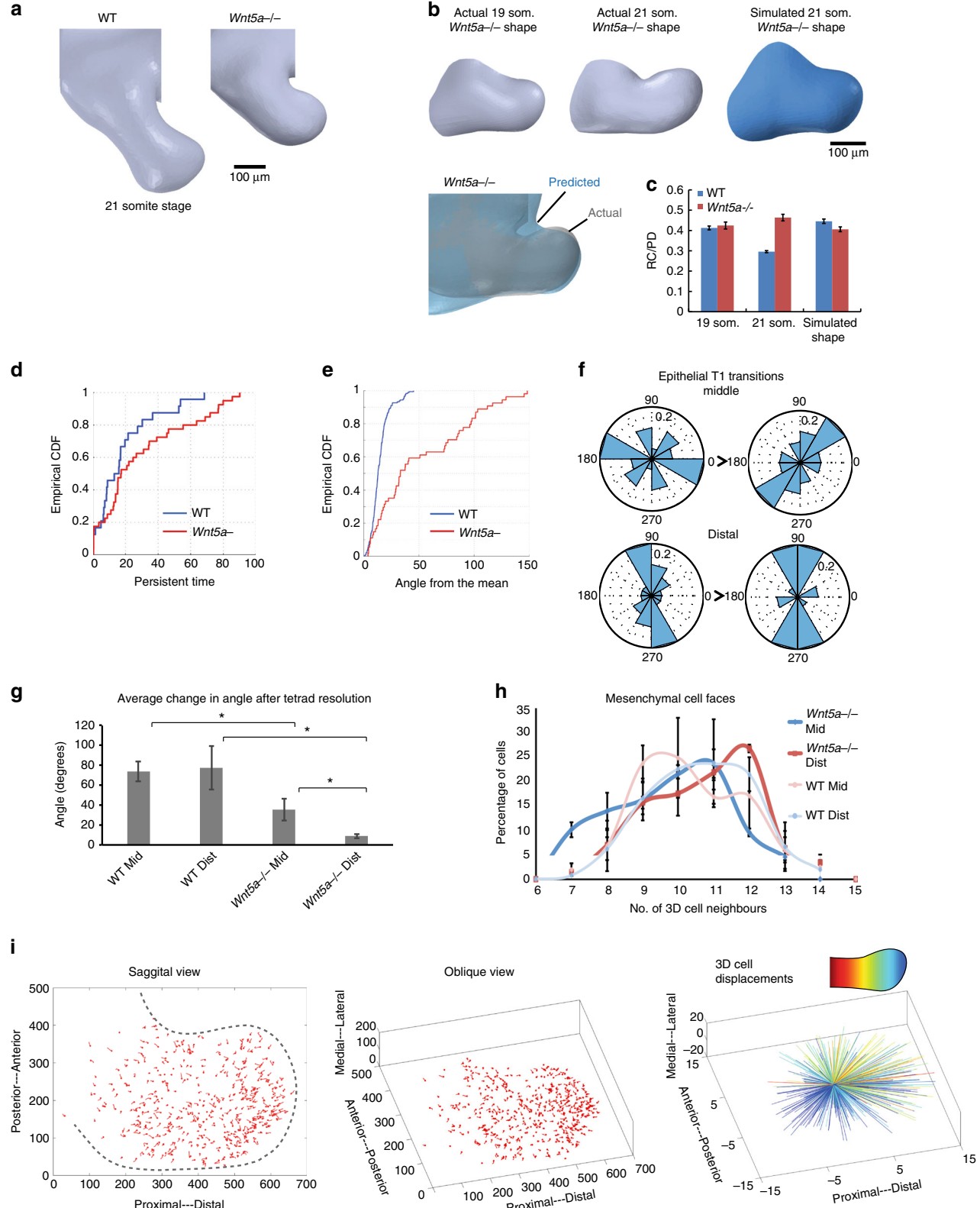

There are interesting parallels between our findings and those observed in a chick model of Robinow syndrome induced by viral expression of human mutant *WNT5A* alleles[42]. In both data sets, mandibular hypoplasia was associated with cell polarity and migration, rather than proliferation, defects and overexpression of WT *WNT5A/Wnt5a* also caused hypoplasia. Although potentially interacting pathways were not clear in the chick model, the implication of YAP/TAZ and PIEZO1 downstream of WNT5A suggests that multiple processes are likely affected by mutations that cause Robinow syndrome. Combining increasingly accurate biophysical approaches with genetics will likely help to define morphogenesis pathways, including those that coordinate cell polarity with physical properties.

**Fig. 5** Deficient cell rearrangements in $Wnt5a^{-/-}$ mutants. **a** The $Wnt5a^{-/-}$ mutant mandibular arch is comparatively short and broad by OPT. **b** A finite element model incorporating the spatial variation of $Wnt5a^{-/-}$ mutant cell cycle time, Young's modulus and viscosity was employed to simulate 4 h of growth starting from the actual 19 somite stage $Wnt5a^{-/-}$ mutant mandibular arch shape compared to the actual 21 somite stage $Wnt5a^{-/-}$ mutant arch. **c** Ratio of rostrocaudal to proximodistal axis length (RC/PD) was comparatively better for finite element simulation of $Wnt5a^{-/-}$ mutant growth than for WT. **d**, **e** Cells of $H2B$-$GFP$ transgenic embryos were tracked by 4D light sheet microscopy and, according to a random walk model, persistence of cell movements (persistent time (**d**)) and direction (angle from the mean (**e**)) were diminished in $Wnt5a^{-/-}$ mutants. CDF, cumulative distribution function. **f** Epithelial T1 transitions in the $Wnt5a^{-/-}$ mutant mandibular arch did not tend to converge and extend the middle region as in WT (compare with Fig. 3b). **g** Angular change in long axis among resolving tetrads was diminished in $Wnt5a^{-/-}$ mutant epithelium ($n$ = lateral arch epithelium in each of 2–5 embryos per condition, asterisks indicate $p < 0.05$, Student's $t$-test). **h** Distribution of mesenchymal cell face numbers was shifted rightward to higher values for $Wnt5a^{-/-}$ mutant mesenchyme ($n = 2$ $Wnt5a^{-/-}$ embryos, 92 middle cells, 146 distal cells, error bars denote s.e.m.). **i** Mesenchymal cells in the mutant middle region lacked the centripetal intercalary movements and longitudinal tissue flow as observed in the WT middle region (compare with Fig. 2f, g). Source data are provided as a Source Data file

## Methods

**Mouse strains**. Analysis was performed using the following mouse strains: $CAG::H2G$-$GFP$[72] (Jackson Laboratory: B6.Cg-Tg(HIST1H2BB/EGFP1Pa/J), mTmG[73] (Jackson Laboratory: Gt(ROSA)26Sortm4(ACTB-tdTomato-EGFP)Luo/J)), $CAG::myr$-$Venus$[74], full length vinculin tension sensor (VinTS) – see below, vinculin TFP control (VinTFP), vinculin tailless control (VinTL), $Wnt5a^{+/-}$[34], $Piezo1^{+/-}$[75]. All mouse lines were outbred to CD1. All animal experiments were performed in accordance with protocols approved by the Animal Care Committee Hospital for Sick Children.

**Vinculin tension sensor knock-in mouse strains**. To target the $Rosa26$ locus, constructs were cloned using the following plasmids: vinculin tension sensor (VinTS) and vinculin tailless (VinTL)[58] (Addgene plasmid#: 26019, 260020), and Ai27 (gift from Hongkui Zeng[76]; Addgene plasmid#: 34630). To generate VinTS targeting vector, Mlu1 sites were introduced upstream of the start codon and downstream of the stop codon in VinTS using PCR. To generate VinTL, an Mlu1 site was introduced upstream of the start codon and a stop codon followed by an Mlu1 site downstream of the Venus sequence. To generate vinculin teal fluorescent protein (VinTFP—FRET donor only), an Mlu1 site was introduced upstream of the start codon and a stop codon followed by an Mlu1 site downstream of the mTFP1 sequence. PCR products were digested with Mlu1 and purified using Qiaex gel purification kit (Qiagen). Reverse primers for generating VinTL and VinTFP were identical, so the constructs were distinguished based on size in an agarose gel following Mlu1 digestion and gel purification and bands were cut out accordingly. The inserted construct in the Ai27 Rosa26 targeting vector was removed with Mlu1 and replaced with VinTS, VinTL, or VinTFP. Sequences for all targeting vectors were confirmed through DNA sequencing (performed by The Centre for Applied Genomics, The Hospital for Sick Children). The final targeting vectors were constructed as follows: CAG enhancer—FRT—loxP—stop codons— 3× SV40 poly(A)—loxP—VinTS/VinTL/VinTFP—WPRE—bGH poly(A)—attB— PGK promoter—FRT—Neo—PGK poly(A)—attP.

Generation of ES cell lines and generation of chimeras were performed by the Transgenic Core at the Toronto Centre for Phenogenomics. Briefly, linearised constructs were electroporated into G4 ES cells and G418-resistent clones were screened by PCR. 4 positive clones from VinTS, and 2 positive clones from each of VinTL and VinTFP were aggregated with CD1 morula to obtain chimeric mice following standard procedures. Chimeric mice were outbred to CD1 mice to obtain an F1 generation through germline transmission. All procedures involving animals were performed in compliance with the Animals for Research Act of Ontario and the Guidelines of the Canadian Council on Animal Care. The Toronto Centre for Phenogenomics (TCP) Animal Care Committee and Hospital for Sick Children Animal Care Committee reviewed and approved all procedures conducted on animals at TCP and at the Hospital for Sick Children, respectively.

**Z/Wnt5a conditional overexpression mouse strain**. The Z/Wnt5a line was generated by targeting a Cre-activated Wnt5a expression vector upstream of the endogenous Ubiquitin-b (Ubb) locus using a similar strategy as described for generating the Z/Norrin expression cassette[77] (mouse strain provided by Phil Smallwood and Jeremy Nathans). Expression of the Wnt5a transgene is activated only after Cre-mediated recombination, which removes a LacZ-STOP cassette upstream of the Wnt5a open reading frame. Sox2:Cre[78] was employed to drive ubiquitous expression.

**Optical projection tomography**. E9.5 mouse embryos were harvested and fixed in 4% paraformaldehyde overnight at 4 °C. OPT was performed using a system that was custom-built[79]. Three-dimensional (3D) data sets were reconstructed from auto-fluorescence projection images acquired over a 25 min scan period at an isotropic voxel size of 4.5 μm. The 3D surface renderings of the OPT data were generated by MATLAB software, version R2011b (Mathworks).

**Cell cycle time measurement**. Pregnant females were injected first with CIdU intraperitoneally at E9.25 and then with IdU after 2.5 h. Thirty minutes following the second injection, embryos were dissected in cold PBS and fixed with 4% PFA

overnight at 4 °C. Whole-mount immunofluorescence against CIdU and IdU was performed[43].

**Elasticity and viscosity measurement by AFM**. Mouse embryos were incubated in 50% rat serum in DMEM on a 35 mm dish in which 2% agarose was poured around the perimeter. The mandibular arch was immobilised to the agarose with pulled glass needles pinned through the flank adjacent to mandibular arch. The arch was examined using an AFM (BioScope Catalyst, Bruker) mounted on an inverted microscope (Nikon Eclipse-Ti). AFM indentation tests were performed using a spherical tip (radius: 15 μm) at distinct locations categorised as proximal, middle and distal mandibular arch. Spherical tips were made by assembling a borosilicate glass microsphere onto a tipless AFM cantilever using epoxy glue. The cantilever spring constant was calibrated before every experiment by measuring power spectral density of thermal noise fluctuation of the unloaded cantilever.

To determine the elastic modulus of epithelium, a trigger force of 300 pN was consistently applied. For both small and large indentation measurements in each region (proximal, middle, and distal), 15 locations were measured and repeated in triplicate, amounting to 45 measurements at each region. Because the epithelial thickness is approximately 10 μm, we followed the empirical 10% rule[80] to indent epithelium up to 1 μm in depth to avoid influence from the mesenchyme. The Hertz model for a spherical tip was used to calculate the elastic modulus of the epithelium from the small indentation data. To determine mesenchymal elastic modulus and the overall tissue viscosity, we applied large indentation (depth: 7–10 μm) at indentation rates of 5, 10 and 15 μm/s. The Kelvin–Voigt model was used to fit the large indentation force–displacement data. To extract overall tissue viscosity, the range of data beyond 1.5 μm indentation depth was used, and the elastic range was neglected. The elastic modulus value of mesenchyme was then determined by using experimental force–displacement data and finite element simulation.

**Determination of epithelial elastic modulus**. For small indentation, the Hertz model for a spherical tip was used to fit the experimental force–displacement data and determine epithelial elastic modulus values. The relationship between the indentation depth $d$ and the loading force $F$ is

$$F = \frac{4}{3} E^* R^{1/2} d^{3/2}, \tag{1}$$

$$E^* = \frac{1 - v_1^2}{E_1} + \frac{1 - v_2^2}{E_2}, \tag{2}$$

where $E_1$, $v_1$ are the elastic modulus and Poisson's ratio of the indenter; and $E_2$, $v_2$ are the elastic modulus and Poisson's ratio of epithelium. The spherical tip was made of borosilicate glass (elastic modulus and Poisson's ratio: 63 Gpa and 0.2). The Poisson's ratio for epithelium was set to 0.4[81]. Using the experimental data from small indentation as well as the above calculation, epithelial elastic modulus $E_2$ was quantified.

**Determination of embryonic tissue viscosity**. For large indentation, the Kelvin–Voigt model was used to fit experimental force–displacement data. Stress–strain relation in the Kelvin–Voigt model is

$$\sigma(t) = E\varepsilon(t) + \eta \frac{d\varepsilon(t)}{t}. \tag{3}$$

When the equation is multiplied by contact area $S = \pi a^2$, where $a$ is the contact radius, the left-hand side results in total force applied to epithelium and mesenchyme. For a bilayer structure, the contact radius $a = f\left(\frac{E_1}{E_2}, t, d, R\right)$ is a function of the first-second layer elastic mismatch ratio $E_1/E_2$, the first layer thickness $t$, indentation depth $d$, indenter radius $R$. Comparing the elastic modulus of epithelium $E_{epithelium}$ and the overall modulus (i.e., epithelium and mesenchyme both included), we found the ratio $E_{epithelium}/E_{mesenchyme}$) was small (~2). Thus, according to the Hertz model[82], $a = \sqrt{dR}$. Rewriting Eq. (3) gives

$$F = F_{elastic} + F_{viscous} = \frac{4}{3} E^* R^{1/2} d^{3/2} + \frac{2}{3} \eta R^{1/2} d^{1/2} v, \tag{4}$$

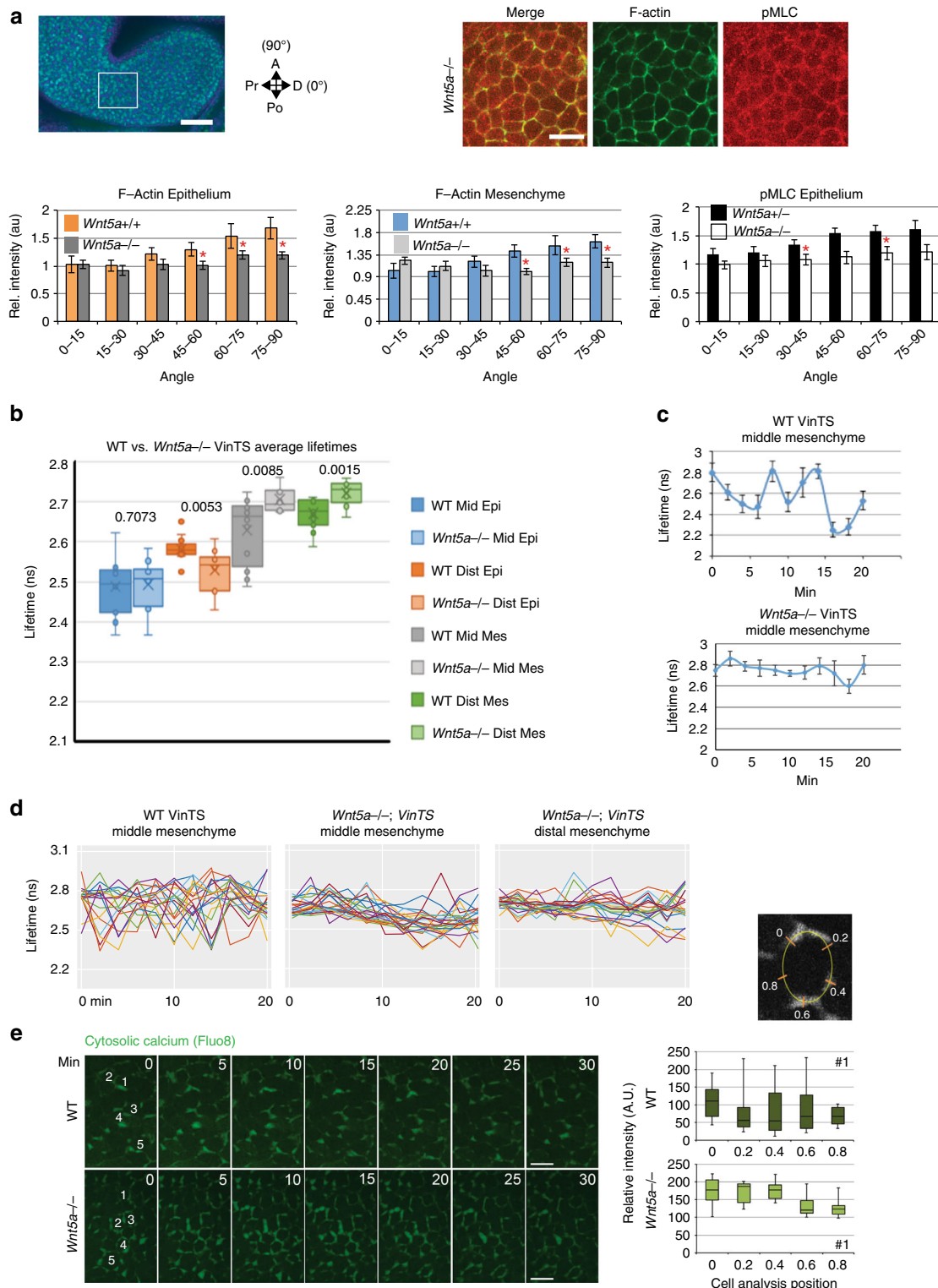

where $\nu$ is indentation rate. It is evident that $F_{elastic}$ is rate-independent while $F_{viscous}$ is rate-dependent. Therefore, $F_{elastic}$ and $F_{viscous}$ were determined with the force-displacement data measured at different indentation rates, and viscosity $\eta$ was also calculated. Furthermore, $F_{elastic}$ was exported into the finite element model to calculate mesenchyme's elastic modulus.

**Determination of mesenchymal elastic modulus.** A snapshot of the meshed 2D axisymmetric finite element model with a bilayer structure (epithelium and mesenchyme) under frictionless loading is shown in Supplementary Fig. 1B. Supplementary Fig. 1F shows the steps we used to conduct iterative FE simulation to quantify mesenchyme's elastic modulus. As in the above section, for each

displacement, $F_{elastic}$ was determined. This elastic force applied to the overall tissue was incorporated into the finite element model. Since epithelial elastic modulus has been determined, different modulus values were assigned to the mesenchymal layer in the finite element model until finite element-obtained force–displacement curves agreed well with the experimental curves (Fig. 1f). Mesenchymal elastic modulus was saved once the $R^2$ value (correlation coefficient between the two curves) was greater than the threshold value (0.99).

**Finite element modelling.** An optical projection tomography (OPT) image stack was first concatenated and reconstructed using ImageJ, and a 3D model was exported in the STL format. We used MeshLab to isolate the mandibular arch

**Fig. 6** Actomyosin bias and vinculin tension were diminished in the *Wnt5a*$^{-/-}$ mutant mandibular arch. **a** Spatial distributions of F-actin and pMLC were not biased in 21 somite stage *Wnt5a*$^{-/-}$ mutants. The angular distribution of immunostain fluorescence intensity for epithelial (*n* = 4 embryos) and mesenchymal (*n* = 4 embryos) F-actin, and epithelial pMLC (*n* = 5 embryos) was quantified relative to the arch long axis that was designated 0° using SIESTA. Scale bars: whole arch panel 100 μm, merge field 20 μm, asterisks denote *p* < 0.05, Student's *t*-test, error bars denote s.e.m. **b** Average fluorescence lifetime values in middle (mid) and distal (dist) epithelium and mesenchyme of mandibular arch. Boxplots show mean (x), median (---), central quartiles (coloured box), and range (end bars); *n* = 15 cells per region x 2 (mutant) or 3 (WT) embryos. *p* values for pairwise WT/*Wnt5a*$^{-/-}$ mutant comparisons are given in the graph. Of note, middle mesenchymal lifetime range was diminished in *Wnt5a*$^{-/-}$ mutants and more closely resembled WT distal mesenchymal lifetime values. **c** Representative vinculin force curves of an individual *Wnt5a*$^{-/-}$ mutant cell in the middle region that lacks the amplitude observed for WT cells in same region. **d** Multiple vinculin force curves; the variance (amplitude) of lifetime values for *Wnt5a*$^{-/-}$ mutant middle mesenchyme (0.0122 ns, *n* = 3 embryos, 15 cells per embryo) was diminished relative to WT middle mesenchyme (0.0201 ns, *n* = 5 embryos, 15 cells per embryo, *p* = 0.04), but similar to WT distal mesenchyme (0.0132 ns, *p* = 0.38, ANOVA). Lifetime variance was similar between WT (0.0132 ns) and *Wnt5a*$^{-/-}$ mutant (0.0096) distal mesenchyme (*p* = 0.13). **e** Variation of cytosolic calcium concentration using Fluo8 applied to embryos was quantified at 5 positions per cell over time. Corresponds to Supplementary Fig. 6D; scale bar 10 μm. Source data are provided as a Source Data file

from the embryo. Quadric edge collapse decimation algorithm was employed to reduce the total number of 3D triangular mesh faces to the target number of faces (~5000 without distorting the geometry). The model was then imported into Solidworks in which the 3D mandibular arch model was segmented into two layers (epithelium and mesenchyme) with 24 regions based on the geometric locations of cell cycle time measurements. The finite element model was fixed in all six degrees of freedom (Displacement: $U_x = U_y = U_z = 0$, Rotation: $U_{Rx} = U_{Ry} = U_{Rz} = 0$) at the extended proximal end to ensure the boundary condition did not influence growth of the proximal region (Supplementary Fig. 1G). A ten-node tetrahedral element (SOLID 186) was selected to discretise the model.

Each region of the model was assumed to be viscoelastic, isotropic and homogeneous. The viscoelastic property of each region was assigned with an average value obtained from experimental AFM measurement and implemented in ANSYS v15.0 (ANSYS Inc., Canonsburg, PA) with instantaneous elasticity and two-pair Prony relaxation. We also performed comparative simulations with different viscoelastic values within the range of our AFM measurement and predicted similar tissue shape change with no significant difference.

The experimentally measured 3D cell cycle time values were converted to a spatial pattern of strain[44], and strain values were incorporated in the finite element model for tissue shape prediction. Cell density was considered to be the number of cells per volume. $d = N/V$. Therefore, tissue volume change was a function of cell number change and cell density change. We started by assuming cell density remains constant (changes in cell density were incorporated later as a correction factor), and tissue volume change was proportional to cell number change $\Delta N/\Delta V = N/V$. Volume strain $\varepsilon_N$ induced by cell number change was thus equal to $\Delta V/V = \Delta N/N$. $\Delta N/N$ was calculated from cell doubling time. Cell density change (measured to be 1.7% per hour) was incorporated into volume strain as a correction factor $\varepsilon_d$. Hence, the overall volume strain $\varepsilon$ was $\varepsilon_N - \varepsilon_d$. The parameters incorporated into the finite element model are shown in Supplementary Table 1.

**Voronoi tessellation and rigidity analysis.** Using light sheet microscopy, we obtained approximately 1000 2-dimensional images (ML-PD plane) of the mandibular arch, each image about 0.4 microns apart from the next along the perpendicular RC axis. Cell nuclei marked by *H2B-GFP* emitted light with relatively high intensity. Each image was processed as following: (a) using a Gaussian deconvolution with a standard deviation roughly the size of one cell, the images were smoothened. (b) Local maxima were sought in square boxes, again, roughly the size of one cell. Each local maximum was marked as a cell centre. This process was repeated for each image in ML-PD plane. The layers of marked cells were then compared against one another. If a point is marked to be the centre of a cell within seven or more consecutive layers in the stack, the middle layer was marked as the RC position of that cell. This process gave us a 3D representation of roughly 6000 cell nuclei in the tissue. Voronoi tessellation was performed using these cell nuclei as nodes for the tessellation algorithm.

**Live time-lapse confocal microscopy.** Live image acquisition was performed by submerging embryos were submerged in 50% rat serum in DMEM without phenol red (Invitrogen) in a 25 mm imaging chamber. Cheese cloth was used to immobilise the embryo and position the mandibular arch directly against the coverglass. Embryos were imaged in a humidified chamber at 37 °C in 5% $CO_2$[7,23]. Time-lapse images were acquired on a Quorum WaveFX-X1 spinning disk confocal system (Quorum Technologies Inc.) at 20× magnification. Images were processed with Volocity software or ImageJ/Fiji. Representative images are shown from at least three independent experiments for each condition, and unless otherwise indicated, from at least three independent cohorts. No statistical method was used to predetermine sample size. Experiments were not randomised. Investigators were not blinded to allocation during experiments and outcome assessment.

**Live time-lapse light sheet microscopy.** Three-dimensional (3D) time-lapse microscopy was performed on a Zeiss Lightsheet Z.1 microscope. Embryos were suspended in a solution of DMEM without phenol red containing 12.5% rat serum and 1% low-melt agarose (Invitrogen) in a glass capillary tube. Once the agarose had solidified, the capillary was submerged into an imaging chamber containing DMEM without phenol red, and the agarose plug was partially extruded from the capillary until the portion containing the embryo was completely outside of the capillary. The temperature of the imaging chamber was maintained at 37 °C with 5% $CO_2$. Images were acquired using a 20×/1.0 objective with dual-side illumination, and a *z*-interval of 0.479 μm that was automatically calcuated based on the numerical aperature (1.0) of the objective.. All experiments were imaged in multiview mode with 3 evenly-spaced views spanning approximately 90 degrees (from a frontal view to a sagittal view of the mandibular arch). Images were acquired for 3–4 h with 10 min intervals. Fluorescent beads (Fluospheres 1 μm, Thermofisher, 1:10$^6$) were used as fiducial markers for 3D reconstruction and to aid in driftcorrection for cell tracking. Multi-view processing was performed with Zen 2014 SP1 software to merge the 3 separate views and generate a single 3-dimensional image. Further analysis and cell tracking were performed using Arivis Vision4D software (Arivis).

**Membrane segmentation and 3D cell neighbour counting.** 3D timelapse datasets of cell membranes were processed with the ImageJ macro Tissue-CellSegmentMovie (kindly provided by Dr. Sébastien Tosi from the Advanced Digital Microscopy Core Facility of the IRB Barcelona) to generate membrane segments prior to analysis with Imaris software (Bitplane). Surface objects were created using Imaris and this data was used for 3D cell neighbour analysis. Cell neighbours were manually counted by identfying the target cell and counting all cells, which share an interface within the plane, as well as adjacent cells in the planes above and below. Analysis was performed on three separate areas in both middle and distal regions of the mandibular arch over two independent experiments for each condition.

**Cell tracking and dandelion plots.** Cell tracks (cell positions tracked over time) were calculated manually. Cell nuclei were followed between image *z*-stacks at 5 or 10 min intervals. The large drift between images over time and the resolution of the images made it difficult to follow more than 130 cells throughout the entire movie. Only cells that could reliably be identified and followed by eye in at least 33 frames were considered for the random walk analysis.

For each movie, the *z*-stacks for each time point were aligned, and fusion stacks were created to generate *z*-plane images based on the mean voxel intensity at each voxel. The fused stacks were then imported into Arivis Vision 4D software (arivis AG, Unterschleißheim, Germany), and automated tracking was used to generate cell tracks. Tracks were validated manually, and validity annotations were entered into an Excel spreadsheet. Tracks were defined to be valid at a given timepoint if the segment determining the track's location at that timepoint was centred in the middle of a nucleus. Tracks were defined to be suitable for use in further analysis if time between their first valid timepoint and final valid timepoint was 150 min or greater. Whether a track tracked an epithelial or mesenchymal cell was determined observing the nuclear location of the track relative to the tissue boundary as well as the morphology of the nucleus (epithelial cells tended to have more elongated, columnar shapes relative to their counterparts in the mesenchyme).

To correct for drift, the frame by frame displacement of red fluorescent beads that were embedded adjacent to the embryo within the agarose plug used in the light sheet microscope was calculated. These displacements were imported into MATLAB for initial track concatenation and drift correction. Track and segment information and validity were acquired on Arivis. The concatenation function returned drift corrected tracks that were filtered to only include valid tracks. Tracks could then be plotted, and end to end displacements calculated and plotted.

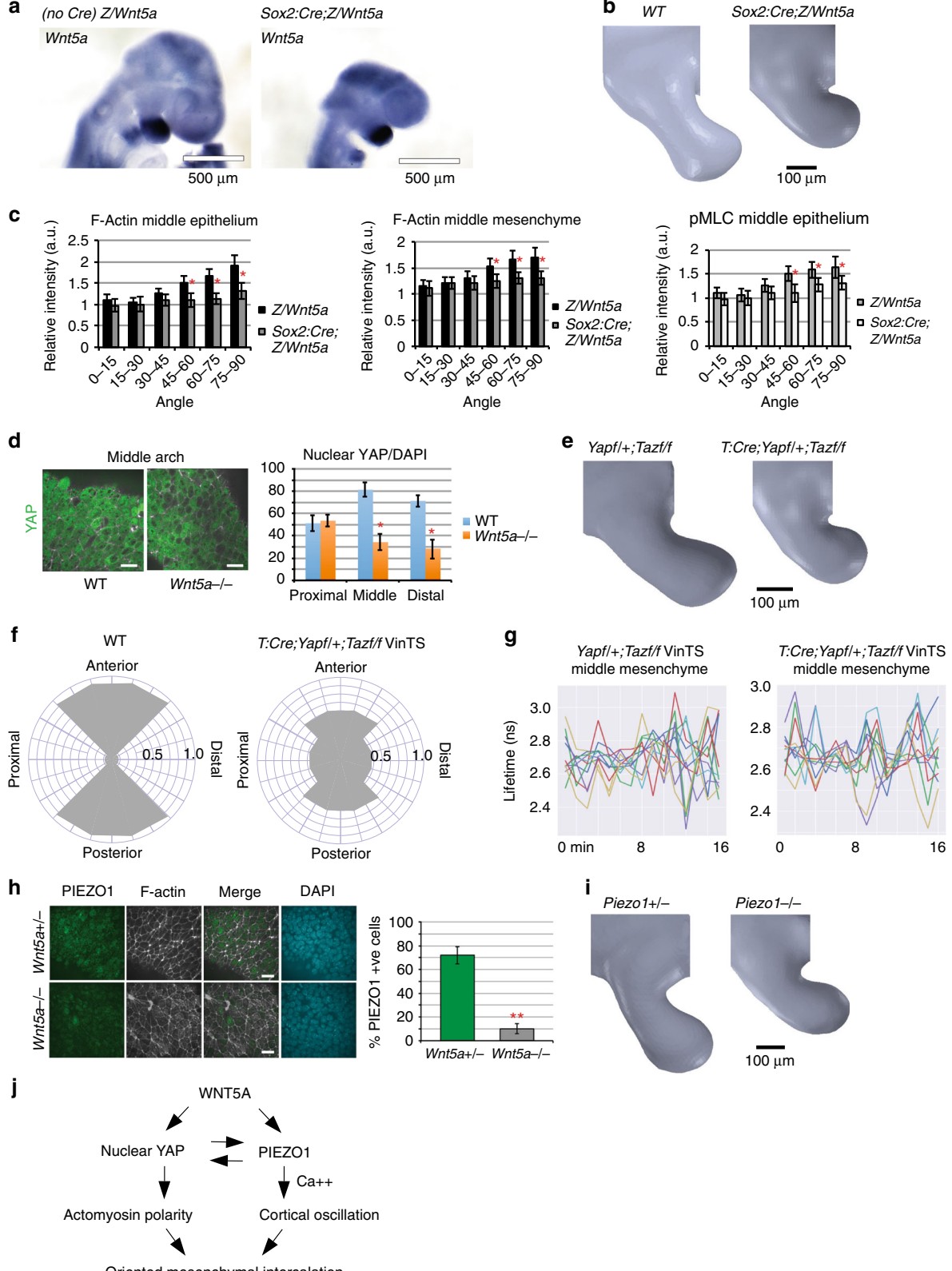

**Random walk model**. We described cell motions from tracking experiments by stochastic processes. In particular, we adopted the Ornstein-Uhlenbeck (OU) process[83] for our analysis. In an OU-process, the trajectory of a particle is determined by a relation describing its change in velocity over time. The change of velocity in the model is proportional to its velocity in the past (persistent) plus a randomly generated term (random walk). This approach has been used to analyse the properties of cell migrations under many contexts including tumour growth[52] and wound repair[51].

The OU-process is defined by the Langevin equation

$$\frac{d\mathbf{v}}{dt} = -\frac{1}{\tau}\mathbf{v} + \frac{\sqrt{2D}}{\tau}\mathbf{w}, \qquad (5)$$

where $v$ is the cell velocity, $t$ is time, $D$ is the diffusion coefficient, vector $w$ describes a Wiener process, and $\tau$ is the time scale often referred to as the persistent time. The persistent time, which may be understood as the length of time a given

**Fig. 7** *Wnt5a* is a spatial cue and *Yap/Taz* and *Piezo1* act downstream of *Wnt5a*. **a** *Sox:Cre;Z/Wnt5a* embryos expressed *Wnt5a* beyond the normal expression domains by in situ hybridisation. **b** OPT showing a short and wide mandibular arch in a 21 somite *Sox:Cre;Z/Wnt5a* embryo. **c** F-actin and pMLC biases were diminished in 20–21 somite *Sox:Cre;Z/Wnt5a* embryos. **d** Proportion of cells exhibiting nuclear YAP in the *Wnt5a$^{-/-}$* mandibular arch. Scale bars: 20 µm. **e** OPT showing phenotype of *T:Cre;Yap$^{f/+}$;Taz$^{f/f}$* mandibular arch. **f** Orientation of mesenchymal cell intercalations in the sagittal plane of mandibular arch mid-regions of WT and *T:Cre;Yap$^{f/+}$;Taz$^{f/f}$* embryos. Intercalation angles were binned into one of 4 arcs of 90° each: anterior, posterior, proximal and distal (*n* for WT = 48, *n* for mutants = 42, where *n* = No. of groups of 5–7 cells). **g** Multiple vinculin force curves showed no difference in the variance (amplitude) of fluorescence lifetime values of *Yap$^{f/+}$;Taz$^{f/f}$* and *Tcre;Yap$^{f/+}$;Taz$^{f/}$* middle mesenchymal cells (p = 0.185, ANOVA, n = 16 cells in each of two 20–21 somite embryos per condition). **h** PIEZO1 immunostain intensity was diminished in the *Wnt5a$^{-/-}$* mandibular arch. Scale bars: 20 µm. **i** The 21 somite *Piezo1$^{-/-}$* mutant mandibular arch partially phenocopied that of *Wnt5a$^{-/-}$* with a short and broad mid-portion. **j** YAP/TAZ and PIEZO1 partially mediate actomyosin bias and cortical oscillation amplitude, respectively, downstream of WNT5A to orient and promote mesenchymal cell intercalations. Source data are provided as a Source Data file

velocity "remembers" itself, describes the time of the velocity auto-correlation function

$$\langle \mathbf{v}(t) \cdot \mathbf{v}(0) \rangle = \frac{nD}{\tau} e^{-t/\tau}, \tag{6}$$

where $n$ is the space dimension of the tracks. The mean squared displacement (MSD) is given by

$$\mathbf{d}(t) = 2nD\tau \left( e^{-\frac{t}{\tau}} + \frac{t}{\tau} - 1 \right) \tag{7}$$

Equations (6), (7) was used to fit the observed cell tracks to obtain the persistent time $\tau$ and the diffusivity coefficient $D$. Consequently, we were able to simulate cell trajectories by Eq. (5) to further understand similarities and discrepancies between the statistics from the observations and the OU-process.

The wild type dataset that we used for the numerical simulation contained $n = 179$ number of cells tracked with a time interval of $t = 5$ min. over a 3 h period. Following Wu et al.[52], we simulated cell trajectories by applying a first-order Euler's scheme to Eq. (5) to obtain

$$d\mathbf{x}(t, dt) = \left( 1 - \frac{dt}{\tau} \right) d\mathbf{x}(t - dt, dt) + \sqrt{\frac{S^2 dt^3}{\tau}} W, \tag{8}$$

where $W \sim N(0, 1)$ and $S$ is the cell speed with $D = S^2\tau$. We applied principle component analysis to observed cell trajectories prior to our analysis so that we could simulate cell displacement in each direction separately. In other words, we diagonalised the correlation coefficient matrix so that off-diagonal components were zero.

An important characteristic of a random walk model is its mean squared distance (MSD). For an arbitrary trajectory, the MSD as a function of time follows a power-law, with power of 0 representing a truly random trajectory, and power of 2 representing movement in a straight line with no randomness. On a log–log plot, these two extremes translate into lines with slope 0, or a horizontal line, for the random trajectory, and a line of slope 2 for the walk in a straight line. A persistent random walk, however, is characterised by slope 1, the short black line in Supplementary Fig. 1g. The analysis of cell trajectories (based on light sheet microscopy) suggested that mandibular arch cells exhibited a MSD slope that is characteristic of a persistent random walk.

**Strain analysis.** A rectangular grid of points was superimposed on the first frame of a rostrocaudal-proximodistal plane of a time-lapse light sheet movie of the mandibular arch. The points were followed in subsequent frames by calculating the correlation function of a box around each point in the initial time frame to all the boxes of the same size in the vicinity of the original box in the next frame. The vector that connected the location of the box in the first frame to the location of the neighbouring box with the highest correlation with the original box in the second frame was the displacement vector. The points were moved by this displacement vector in each time frame, and this process was repeated for all 31 time frames.

**Immunostaining.** Embryonic day (E) 9.0–9.5 mouse embryos were fixed overnight in 4% paraformaldehyde in PBS followed by three washes in PBS. Embryos were permeabilised in 0.1% Triton X-100 in PBS for 20 min and blocked in 5% normal donkey serum (in 0.05% Triton X-100 in PBS) for 1 h. Embryos were incubated in primary antibody overnight incubation at 4 °C. Embryos were washed in 0.05% Triton X-100 in PBS (4 washes, 20 min each), and then incubated in secondary antibody (1:1000) for 1 h at room temperature. Embryos were washed (4 washes, 20 min each), followed by a final wash overnight at 4 °C, and stored in PBS. Images were acquired using a Quorum spinning Disk confocal microscope (Zeiss) at 10×, 20×, or 40× magnification, and image analysis was performed using Velocity software (Perkin Elmer) and ImageJ.

**Antibodies.** Phospho-myosin light chain 2 (Thr18/Ser19) (Cell Signalling #3671, rabbit, 1:250); PIEZO1 (FAM38A, proteintech #15939–1-AP, rabbit, 1:250); N-cadherin (BD #51–9001943, mouse 1:250); Desmoglein (BD #51–9001952, mouse, 1:250); CIdU (Abcam ab6326, rat, 1:250); Idu (BD Biosciences 347580,

mouse, 1:250); Calpain2 (E-10) (Santa Cruz #sc-373966, mouse, 1:50); VANGL2 (Sigma HPA027043, rabbit, 1:250). All secondary antibodies were purchased from Jackson Immunoresearch and used at 1:1,000 dilutions. Rhodamine phalloidin (Invitrogen #R415, 1:1,000); Alexa Fluor 633 phalloidin (Invitrogen #A22284, 1:1000).

**Whole mount in situ hybridisation.** Whole mount in situ hybridisation was performed after fixation in 4% paraformaldehyde followed by dehdration through a methanol series. Embryos were bleached then underwent digestion in proteinase K, hybridisation with probe in formamide, SSC, SDS and heparin, treatment with anti-Digoxygenin-AP, and colourimetric reaction[84]. Wildtype and mutant littermate embryos were processed identically in the same assay for comparison. The *Wnt5a* riboprobe was gift from A. P. McMahon.

**Quantification of polarised actin and myosin distribution.** Single confocal slices taken 2 µm above (for epithelium) and below (for mesenchyme) the basal surface of epithelial cells stained with phospho-myosin light chain 2 (Thr18/Ser19) (pMLC), rhodamine–phalloidin or Alexa Fluor 633 phalloidin were analysed using SIESTA software[7,85]. Cell interfaces were manually identified, and average fluorescence intensities were calculated for all interfaces and grouped into 15° angular bins, with 0°–15° bin representing interfaces that are parallel with the AP axis (PD interfaces) and 75°–90° bin representing interfaces that are parallel with the PD axis (AP interfaces). Average fluorescence intensity values for each bin were normalised to average fluorescence intensity of PD interfaces (0°–15° angular bin). Error bars indicate standard error of the mean and *p* values were calculated using Student's *t*-test.

**Quantification of cell behaviours.** Metaphase-to-telophase transition angles were measured using a line between the centres of daughter nuclei with respect to a proximodistal reference axis[23]. Epithelial tetrad formation and resolution angles were measured by first assigning a proximodistal reference axis taken from a low 10× confocal magnification view of the embryo flank. Tetrads were identified manually, frame by frame. The angle between the long axis of the ellipse outlined by each tetrad and the reference axis was documented at the beginning of a given movie and upon resolution.

**Live time-lapse fluorescence lifetime analyses.** Fluorescence lifetime microscopy (FLIM) was performed on a Nikon A1R Si laser scanning confocal microscope equipped with PicoHarp 300 TCSPC module and a 440 nm pulsed diode laser (Picoquant). Data were acquired with a 40×/1.25 water immersion objective with a pixel dwell time of 12.1 µs/pixel, 512 × 512 resolution, and a repetition rate of 20 MHz. Fluorescence emission of mTFP1 was collected through a 482/35 bandpass filter. Embryos were prepared in 50/50 rat serum/DMEM for live time-lapse microscopy and imaged in a humidified chamber at 37 °C in 5% $CO_2$. Data were acquired over 15 frames at 2 min intervals for 20 min.

Fluorescence lifetime of mTFP1 was determined using $n$-exponential reconvolution model in SymphoTime software (Picoquant) with model parameters $n = 2$ for VinTS and VinTL and $n = 1$ for VinTFP. Fitting was performed to achieve a chi-squared of $\chi^2 = 1.000 \pm 0.100$.

FRET efficiency (E) was calculated based on the molecular separation between donor and acceptor and the corresponding fluorescence lifetime using the following equation:

$$E = R_0^6 / \left( R_0^6 + R^6 \right) = 1 - (\tau_{DA} / \tau_D), \tag{9}$$

where $R_0$ is the Förster radius of donor-acceptor pair, $R$ the donor to acceptor separation distance, TDA the fluorescence lifetime of donor in the presence of acceptor, and TD the fluorescence lifetime of donor only.

Force was calculated from FRET efficiency based on the calibration results reported by Grashoff et al.[58].

**Preparation of embryoid bodies**. Embryonic stem (ES) cell clones of VinTS, VinTL, VinTFP knock-ins were generated by the Transgenic Core at The Centre for Phenogenomics and cultured using standard protocols[86]. Cells were cultured in DMEM (high glucose 4500 mg/L, Invitrogen) supplemented with 2 mM GlutaMAX (Invitrogen), 0.15 mM monothioglycerol, 0.1 mM MEM Non-essential amino acids, 1 mM sodium pyruvate, 50 μg/mL Penicillin/Streptomycin, 1000 U/mL Leukaemia Inhibitory Factor (Chemicon), 15% foetal bovine serum (ES cell qualified, Gibco) on a feeder layer of mouse embryonic fibroblasts. Prior to transfection, ES cells were removed from the feeder layer and cultured in gelatin-coated plates for 2 passages. Cells were transfected with Cre recombinase using Lipofectamine 2000, according to the manufacturer's protocol. YFP-positive cells were sorted by flow cytometry, which was performed in the SickKids Flow Cytometry Facility. Positive cells were recovered and grown on gelatin.

To differentiate cells into embryoid bodies (EBs), cells were diluted in differentiation media (ES cell culture media as described above, except with 5% FBS and without LIF and non-essential amino acids) at a concentration of 1000 cells per 10 μL and plated in drops of 20 μL on the lid of a 10 cm plate. Cells were grown in this hanging drop configuration for 6 days to allow for embryoid body differentiation. EBs were harvested and fluorescence lifetime was measured using the LIFA system by Lambert Instruments on an Olympus IX81 equipped with an Li2CAM iCCD with GenIII GaAs intensifier. Spontaneous differentiation of EBs to contracting myocardial cells continued in 24-well plates[87].

**Inhibitor treatments**. The effect of inhibitors Y27632 (Sigma #Y0503) and Cytochalasin D on fluorescence lifetime of VinTS were conducted as follows: Embryos were incubated with 20 μM Y27632 or 20 μg/mL Cytochalasin D for 15 min in 50% rat serum with DMEM in roller culture with 40% $O_2$, 5% $CO_2$ at 37 °C prior to FLIM experiments that were conducted using the same conditions as above.

**Cytosolic calcium fluctuation**. Calcium indicators Fluo-8 AM (Abcam #142773) or X-rhod-1 AM (Invitrogen #X14210) at 10 μM were added in 2 mL of 50% rat serum medium and incubated for 30 min at 37 °C. Embryos was permiabilised with 0.1% Pluronic F-127 (Invitrogen #P3000MP) in culture medium for 20 min before X-rhod-1 staining. The culture medium was removed and washed (two times) and replaced with 2 mL culture medium. Live cell calcium imaging was performed on a Quorum Spinning Disk confocal microscope (Zeiss) equipped with a 40× water objective lens or a Nikon A1 confocal microscope (Nikon) equipped with a 20× dry objective lens. Calcium indicators were excited with an argon laser line (488 nm), and emissions were recorded in the green channel (500–560 nm) for Fluo-8 AM (acetoxyomethyl) and red channel (600–700 nm) for X-rhod-1 AM. Images were acquired from a single confocal plane every 2 or 5 min for up to 20 min; image and data acquisition was performed using Volocity (Perkin Elmer) or Image J.

**VinTS/Ca$^{2+}$ correlation**. We calculated the correlation between VinTS lifetime and $Ca^{2+}$ concentration data series with zero time-lag. In this study, we defined the percentage of X-rhod-1 AM staining area for each cell in relation to $Ca^{2+}$ concentration. Assume the first time series (force vs. time) are called $x$ ($x1$, $x2$, $x3$, etc. for time points 1, 2, 3), and the second series are $y$ ($Ca^{2+}$ concentration vs. time). First we calculated the mean and standard deviation of $x$ and $y$. Let us refer to the mean of $x$, $\mu_x$ and standard deviation of $x$ as $\sigma_x$. $\mu_y$ and $\sigma_y$ for $y$. $N$ is the number of data points. We subtracted the means from each series and multiplied data points corresponding to the same time, and summed the results. The formula follows:

$$\frac{1}{N}\frac{1}{\sigma_x \sigma_y}\sum_{i=1}^{N}\left(x_i - \mu_x\right)\left(y_i - \mu_y\right) \tag{10}$$

We analysed the amplitude of normalised correlation by using MATLAB xcorr (https://www.mathworks.com/help/signal/ref/xcorr.html).

This will give us a number between −1 and 1. For values approaching 1, the two series are strongly correlated. Approaching 0, they are not correlated. Approaching −1 implies they are anti-correlated.

**Reporting summary**. Further information on experimental design is available in the Nature Research Reporting Summary linked to this article.

## Data availability

The authors declare that all data supporting the findings of this study are available within the article and its supplementary information files or from the corresponding author upon reasonable request. The source data underlying Figs. 1c, d, f, g; 2b,e; 3e; 4c, f; 5c, g, h; 6a, b, e and 7c, d, h and Supplementary Figs. 3d; 4b, c,d, f; 5a–d; 6a–d, and 7a, c–g are provided as a Source Data file.

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

## Acknowledgements

We thank Phil Smallwood and Jeremy Nathans for the *Z/Wnt5a* mouse strain, Matthias Merkel, Lisa Manning and Rodrigo Fernandez-Gonzalez for helpful discussions and reviews of the manuscript. This study was supported by the Grace Bowen Tribute Fund and the CIHR (MOP 126115) to S.H. and by Canada Research Chairs to Y.S.

## Author contributions

Biological question, experimental design: H.T., M.Z., K.L., R.A., Y.S.; S.H.; Cell cycle times, immunostains, actomyosin polarity, Ca$^{++}$ fluctuation: H.T.; OPT.: S.S., M.H.; Atomic force microscopy, finite element modelling: M.Z., X.W., Y.S.; 4D cell tracking: O.W., M.Z.; Vinculin tension sensor knock-in and measurements in vivo, light sheet imaging: K.L., H.T.; Vinculin tension sensor ceiling/floor assessments in vitro/in vivo: K.L., N.H., W.L., M.V., D.H.; Vinculin tension sensor anisotropy, group effects: K.F., C.P.; Vinculin tension sensor oscillatory and FRET analysis: S.G., A.D.; Random walk model: X.X., H.H.; Strain and rigidity analyses: M.S., H.H.; Epithelial cell rearrangement analysis: X.X.C., M.W.; Mesenchymal cell rearrangement analysis: S.H.; Generation of Wnt5a overexpressing transgenics: E.K., H-Y.H.; Optical projection tomography: S.S., M.H.; VANGL2 evaluation: H.T., J.F., R.A.; Vinculin tension sensor advice: A.D.; Mutant analyses: H.T., K.L., M.Z., J.W.; Manuscript preparation: H.T., M.Z., Y.S., S.H.

## Additional information

**Competing interests:** The authors declare no competing interests.

