## [Peer Review File · Nature Communications]

Reviewers' Comments:

Reviewer #1:

Remarks to the Author:

The manuscript by Tao et al demonstrates Wnt5a mediated 3D mesenchymal cell intercalations are essential for mouse embryo mandibular arch shaping. More interestingly, YAP/TAZ and PIEZO1 are two downstream targets of Wnt5a in this shaping regulation. There are, however, some weak points that need additional experiments or revision as noted below.

Comments:

In the model proposed by the authors, it seems that YAP/TAZ pathway and PIEZO1 pathway are two independent pathways in the Wnt5a mediated arch formation. However, whether there is any cross talk between these two different pathways is not clear. It would be very interesting to check:

- 1) Endogenous YAP staining and nuclear YAP/DAPI ratio in both Piezo1^{+/-} and Piezo 1^{-/-} embryos;
- 2) Endogenous PIEZO1 staining, the analysis of PIEZO1 positive cell percentage and cytoplasmic calcium dynamic analysis in YAP/TAZ excised embryos.
- 3) There is a reported PIEZO1 inhibitor-GsMTx4 (Bae et al Biochemistry 2011). Could PIEZO1 inhibitor treated YAP/TAZ excised embryos completely mimic Wnt 5a^{-/-} embryos in the mandibular arch formation?
- 4) The authors also mention about ROR2, which an important tyrosine kinase in Wnt 5a regulation. How does the mandibular arch grow in ROR2^{-/-} embryos?

Minor comments:

- 1) The asterisks in figure 1 F are very confusing. For example, in proximal group, why 19 Som. Mes group but not 21 Som. Epi group is statistically higher than 19 Som. Epi group? Please provide the exact p-values.
- 2) In supplementary figure 4 D, are the bars representing the average value of lifetimes ranges in different conditions? This way of presenting is very confusing. Plotting lifetime range distribution in histogram will be clearer.
- 3) Please include the Fat4 data mentioned in page 11 paragraph 1. This data will strengthen the point that Wnt 5a is the primary regulator of YAP/TAZ pathway in mandibular shape formation but not Hippo pathway.

Reviewer #2:

Remarks to the Author:

This manuscript investigates mandible development, morphogenesis, and acquisition of shape. The paper is well written and the data are convincing. Multiple interesting imaging modalities along with interesting methods to study tissue biomechanical properties are used. Multiple genetic models are also used.

In general the manuscript is well written and the data are convincing.

The manuscript is highly technical and as a way to make this paper more accessible I would suggest that the authors better define the jargon that is in the manuscript.

Reviewer #3:

Remarks to the Author:

Hirota et al. propose a new mechanism to explain the morphogenesis of the mandibular arch in the developing mouse embryo. They analyze and model the contribution of cell divisions and tissue

elasticity in shaping the arch's bud-like protrusion that forms over 4 hours from the 19 to 21 somite stage. Using a combination of live imaging, modeling, and knock-out analysis they propose that cell intercalations are primarily responsible for shaping the arch, and that these cell behaviors are in part coordinated by the influence Wnt5a, Yap/Taz, and Piezo1.

Their theory proposes an intriguing and important (yet often elusive) link between genetic and mechanical forces. However, despite this intrigue the data presented do not strongly support their conclusions, or in some instances directly contradict statements made either due to a lack of statistical analysis or a small sample size. There is a considerable need for improvements before the reader can be convinced of their claims.

1) The use of statistical analysis throughout the paper is haphazard. In some cases, sample size, standard deviations and p-values are calculated and reported, yet in others there is at best a lack of statistical analysis or at worst the reader is told to make a comparison between different conditions in different figures themselves, and in many cases either a conclusion cannot be drawn or what can does not support that of the author's.

a. For example, Supplemental Figure 7C claims that Piezo1 $-/-$ cells have lower levels of calcium flux compared to WT, but this is not clear at all from the graphs. The sample size is far too low to make any sense out of the large error bars in Fig 6E or Supp. Fig. 6D for the WT condition, and looks otherwise identical to the Piezo1 $-/-$ mutants. Why do the authors claim that Wnt5a $-/-$ mutants show a similar level of calcium flux as WT, yet a dampening in Piezo1 $-/-$ mutants? This is a core conclusion in their paper, that is absolutely not convincing from the data that is presented. Furthermore, it is not clear what Movie 29 is supposed to demonstrate. There is some apparent photo-bleaching of the signal, but how is the reader supposed to determine what exactly is being "dampened"? Side-by-side and careful comparisons between WT and mutant cells would help the reader, as well as a larger sample size.

b. Similarly, in Figure 4:

- ♣ Variation in vinculin force curves between distal and middle mesenchyme is not convincing, a couple of outliers or variation in experiments/embryos in the middle mesenchyme VinTS could be biasing the outcome, and visually is not convincing. This is also true of Figure 6.

- ♣ Too few samples for the tailless and donor only plots to comment

- ♣ Additional questions: was the ratio between VinTS and X-rhod-1 AM staining determined for the entire cell or just for a single plane?

- ♣ In Supplementary Fig 4 the authors claim that VinTS lifetime dampened in presence of Y27632 from distal to mid ectoderm, but this does not appear to be not statistically significant at least for the distal ectoderm, it is slightly more convincing for Cyto D, but again the effect appears small: direct comparisons need to be made between WT and treated/mutant embryos, and not ask the reader to compare two separate figures and make a statistical conclusion.

c. In general, many effects are small, have too small of a sample size, or are not quantified.

2) There is a lack of scale bars and time bars on many figures and movies. Additionally, many figures and movies are too small to clearly see what is happening.

3) The authors use light-sheet imaging to visualize the development of the mandibular arch over a 4 hour period

a. The culture conditions in the Z1 are different than other imaging/incubation conditions, and while the imaging is only done for a relatively short time the impact of this change in conditions on "normal" development is not demonstrated (lower serum, changes in osmolarity, effect of agarose?).

b. Embedding a growing tissue in even 1% agarose has the potential to impose a mechanical constraint on the developing arch, and given the author's attempts to determine a mechanical component in arch morphogenesis it is important to demonstrate this embedding does not interfere with normal growth. Ideally, if possible, embedding only a portion of the arch in agarose or removing the agarose from just around the arch before placing it in the imaging chamber would eliminate the potential for concern.

c. A z-step size of 0.479 microns is quite a bit of oversampling given a likely light-sheet thickness of 2-4 microns on the Z1 for a sample of that size. A larger step size would help to spare the tissue

from excessive illumination.

d. The small field of view in many movies makes it difficult to interpret what is happening. Many of the labels are far too small to read.

e. Without a clear membrane label interpreting intercalation events based on nuclear position alone can be misleading, however given the difficulty in segmenting membranes in crowded environments (particularly with the mT/mG reporter) this is a challenging problem to address. The authors do however briefly show the use of a mosaic myr-Venus reporter that looked fairly nice for visualizing cell membranes/intercalations. More extensive use of this reporter might help to better elucidate/visualize cell behaviors.

f. A higher number of tracked nuclei over a larger time-scale would be beneficial to support the author's conclusions about cell behavior, particularly with regards to intercalation behavior (which is not clear from the small number of nuclei sampled or from the videos). There are a couple registration/drift compensation modules that might be of use to the authors and improve their ability to track cells, particularly since they include beads in their agarose mounting. A free and relatively well-documented example would be the Preibisch multi-view registration plugin for Fiji.

4) Additional questions/comments:

a. It could be interesting if one were to block cell divisions in the arch and compare the resulting shape WT to fully elucidate the contribution of cell proliferation to arch shape.

b. Can the model predict shapes based on cell movements?

c. In several of the mutant OPT images there appears to be at least some narrowing in the waist. If this is due to intercalation as the author's propose, what is the explanation for this narrowing in mutant cases?

Reviewer #4:

Remarks to the Author:

Development of the mammalian mandibular arch during embryogenesis is critical for multiple aspects of lower face development. Abnormalities in this developmental process are associated with a number of human craniofacial birth defects. Although we have a broad understanding of the genetic interactions responsible for shaping the mandibular arch, our understanding of cell behaviour behind these morphogenetic changes in the shape of the arch over time are rudimentary at best. This manuscript uses a variety of approaches to tease apart the cellular properties that lead to the arch having a different shape in its mid region – which is narrower – compared to the distal end, which is more bulbous, during a specific developmental period. The authors find that changes in the rate of cell division, elasticity and viscosity of the tissue alone cannot account for these shape changes. Following on from these observations they determined that there were significant changes in orientated cell movement, cell contacts and cell intercalation behaviors between the mid and distal regions that appear to account for the shape changes. The authors subsequently develop a hypothesis based on cell intercalation behavior and test this idea using a series of mouse mutants. They use elegant molecular genetic approaches to integrate a fluorescence indicator of cortical tension into the mouse embryo and record changes in tension and calcium fluctuations in the different regions of the arch that correlate with the shape differences. Finally, the authors utilize available mouse mutants to link non-canonical Wnt5a signaling through Yap and Piezo to these cell behaviours.

Overall, this is a comprehensive analysis of a specific issue regarding facial morphogenesis that provides an important contribution to our understanding of this process. The study is supported by strong approaches using both genetics and cell biological methodology and the data shown in the main and supplementary figures are for the most part convincing and are supported by extensive movies.

However, there are several problems with the manuscript that warrant attention before it can be considered for publication.

1.

The Discussion is quite brief. If possible the Discussion should place the results found for the mandibular arch in the context of the genetics of human Robinow Syndrome, which can be caused by WNT5A mutations inherited in an autosomal dominant manner. In this respect, the authors should refer to a recent manuscript from Hosseini-Farahabadi et al (JDR, vol 96, pp1265-72 (2017)) that examines aspects of Wnt5a action on mandibular development in the chick. It would also be useful if the data from the current manuscript were placed into a broader context of mouse mutations that cause similar shape changes in the mandibular arch.

2.

Similarly, the discussion should include details of how the studies on Wnt5a described here for the mandible correlate with and inform other biological processes regulated by Wnt5a including limb bud morphogenesis, axial extension, and potentially tumorigenesis to place these studies in a wider context for this gene's role in development, morphogenesis and disease.

3. Results, Methods, Legend and Supp Fig 1. There are three issues regarding cell cycle analysis.

The schematic in Fig Supp 1A does not match the text presented on page 14/15 of the methods either in the order of the injections or the timing between them. This issue needs to be clarified.

The authors also use both IddU and IdU throughout the manuscript interchangeably. I think the latter is the correct term.

The immunological reagent used to detect IdU needs to be listed in the methods section on page 21.

4.

Results. The text on page 8 discussing the development of the VinTS, VinTFP, and vinTL would benefit from a short description of how vinculin would provide a readout of tension, particularly with respect to the presence or absence of the tail domain.

5.

The data concerning the calcium signal fluctuations are not as convincing as presented as other studies in the manuscript, and the meaning of "dampening" used in the context of these experiments could be clearer. For example, in Supp Fig 4G, the data are presented as area, but it is not clear if the dampening represents a change in amplitude alone or also reflects that the overall signal intensity is less in the distal mesenchyme.

6.

With respect to the modeling and morphometric measurements in the control and mutants, which employ cell cycle, elasticity and viscosity measurements, the authors should mention whether the volume of the arch remains constant or whether changes are observed in this parameter that could also impact the analysis of underlying cell behaviours.

7.

The methods section does not contain descriptions of the studies in Supp Fig 4 on the embryoid bodies and cardiomyocytes.

8.

Page 10 and Figure 7A.

Further information should be provided concerning the expression of the Sox2:Cre driven Wnt5a construct. The authors say that Wnt5a becomes ubiquitously expressed and disrupts the gradient in the mandible. However, the expression of the endogenous gene seems to be much higher than the transgene from the in situ analysis. It would be helpful if the authors could quantitate how

much the relative expression of Wnt5a has changed in the proximal mandibular regions in controls versus transgenics. It would be particularly helpful to know how the increased levels of transgene expression in the embryo as a whole relate to the levels of the endogenous gene expression in the distal mandible (high signal) so that it becomes clearer how the gradient has been affected.

Minor Points

Page 11.

"In Wnt5a mutants, nuclear localization of YAP", since this sentence comes soon after the discussion of the Wnt5a overexpressing mice, it would be helpful to say "In Wnt5a loss of function mutants" to clarify the strain under discussion.

Page 20.

"A persistent random walk, however, is characterized by slope 1, the short black line in the figure". Please indicate the figure to which this refers.

Page 31.

Figure 2 and legend. The meaning of the different colours in panels A, H and I should be included. For example, "Coloured lines correspond to components of the strain tensor" does not provide sufficient information concerning the nature of each line shown. The authors could also refer to a relevant movie in the appropriate place in the legend to strengthen the arguments.

Page 31. Supp Fig 2 and legend.

The meaning of the colours in panel H should be explicitly stated.

Page 31, Figure 3 and legend.

A time series is mentioned for panel 3, but no time parameters are shown.

Fig. 4, Fig Supp 4 and Fig 6.

I am unclear why the "Lifetime range (max-min) in vivo" is under 1ns in Supp Fig 4D, but the Average lifetime is over 2ns in other figures and panels.

Page 33, Supp Fig 5D Legend.

"In contrast to WT embryos (see Fig. 1F) middle arch epithelial stiffness did not increase significantly". This should possibly be rephrased to "increase as significantly" since they have asterisks on the figure to show that $P < 0.05$ for these measurements.

Page 35, Supp Fig 7C legend.

The meaning of the individual panels #1, #2 and #3 at the bottom of the figure is not given.

Page 37, Movie 25.

The legend refers to Fig 5E and Supp Fig 5D, but these are not the correct figures.

Typos:

Page 13: "Animal care Committ,ee"

Page 15: "The Possion's ratio"

Page 31: "According to a the model" and "tranjectories"

Supp Fig 2, panel G "Tima lag"

Page 33, Supp Fig 5 Legend. "Between 19 and 20 somite stages" Should be between 19 and 21.

Reviewers' comments:

Reviewer #1 (Remarks to the Author):

The manuscript by Tao et al demonstrates Wnt5a mediated 3D mesenchymal cell intercalations are essential for mouse embryo mandibular arch shaping. More interestingly, YAP/TAZ and PIEZO1 are two downstream targets of Wnt5a in this shaping regulation. There are, however, some weak points that need additional experiments or revision as noted below.

Comments:

In the model proposed by the authors, it seems that YAP/TAZ pathway and PIEZO1 pathway are two independent pathways in the Wnt5a mediated arch formation. However, whether there is any cross talk between these two different pathways is not clear. It would be very interesting to check:

1) Endogenous YAP staining and nuclear YAP/DAPI ratio in both *Piezo1*^{+/-} and *Piezo1*^{-/-} embryos;

Thank you; we performed immunostaining as suggested and show diminished nuclear YAP in *Piezo1*^{-/-} mutants compared to *Piezo1*^{+/-} and *Piezo1*^{+/+} controls in Supp. Fig. 7G. We also show that F-actin bias is lost in *Piezo1*^{-/-} mutants in Supp. Fig. 7H.

2) Endogenous PIEZO1 staining, the analysis of PIEZO1 positive cell percentage and cytoplasmic calcium dynamic analysis in YAP/TAZ excised embryos.

We now show that PIEZO1 is diminished in *Yap/Taz* mutants in Supp. Fig. 7F, and the calcium analysis is shown in Supp. Fig. 7D. These data were useful because they show PIEZO1 and YAP/TAZ positively regulate one another as well as cell polarity and cortical oscillation functions that we regard as central for cell intercalations. The more extensive crosstalk is noted in a slightly revised model in Fig. 7J.

3) There is a reported PIEZO1 inhibitor-GsMTx4 (Bae et al Biochemistry 2011). Could PIEZO1 inhibitor treated YAP/TAZ excised embryos completely mimic Wnt 5a^{-/-} embryos in the mandibular arch formation?

This would have been a great addition, but I'm afraid that GsMTx4 treatment of control and *Yap/Taz* mutant embryos in roller culture resulted in pyknotic nuclei, a sign of cell necrosis, at different concentrations and durations of exposure. Likely as a consequence, we observed very bright calcium reporter signal within cytoplasmic and nuclear compartments, but could not reliably record fluctuation. We could show that *Yap/Taz* mutants were more sensitive to GsMTx4 than WT embryos by examining disorientation of actin and N-cadherin, but the results are not informative given the necrosis.

4) The authors also mention about ROR2, which an important tyrosine kinase in Wnt 5a regulation. How does the mandibular arch grow in *ROR2*^{-/-} embryos?

We now cite four papers in the introduction that clearly document a short and broad mandibular arch in *Ror2* mutant embryos and fetuses that is similar to the *Wnt5a* mutant phenotype.

Minor comments:

1) The asterisks in figure 1 F are very confusing. For example, in proximal group, why 19 Som. Mes group but not 21 Som. Epi group is statistically higher than 19 Som. Epi group? Please provide the exact p-values.

We agree the presentation of statistical significance in Fig. 1F was confusing, thank you. In the revised figure, we've added individual asterisks above bars in the graphs to denote their difference from adjacent bars, and this seems more intuitive. Rather than state every individual p value which would crowd the figure legend, we have provided a range within which all p values fall (10^{-6} - 10^{-19}). We treated Supp. Fig. 5 in the same way. If the reviewer disagrees, we will write every p value.

We removed the lines with asterisks that denoted differences between nonadjacent bars (that spanned epithelial and mesenchymal data) because the potential implications of stiffness differences between the two tissue layers is not addressed.

2) In supplementary figure 4 D, are the bars representing the average value of lifetimes ranges in different

conditions? This way of presenting is very confusing. Plotting lifetime range distribution in histogram will be clearer.

Yes, Supp. Fig. 4D is a complement to Fig. 4C that underscores differences in the dynamic range of lifetime values observed with the full length sensor strain compared to the two control strains. We changed the graph in question to a boxplot that we hope is clearer.

3) Please include the *Fat4* data mentioned in page 11 paragraph 1. This data will strengthen the point that *Wnt5a* is the primary regulator of YAP/TAZ pathway in mandibular shape formation but not Hippo pathway.

Thank you; we now show the *Fat4* mutant embryo heads in Supp. Fig. 7B.

Reviewer #2 (Remarks to the Author):

This manuscript investigates mandible development, morphogenesis, and acquisition of shape. The paper is well written and the data are convincing. Multiple interesting imaging modalities along with interesting methods to study tissue biomechanical properties are used. Multiple genetic models are also used.

In general the manuscript is well written and the data are convincing. The manuscript is highly technical and as a way to make this paper more accessible I would suggest that the authors better define the jargon that is in the manuscript.

Thank you; we have revised many parts of the text and legends to clarify confusing or poorly written sections and to diminish jargon.

Reviewer #3 (Remarks to the Author):

Hirota et al. propose a new mechanism to explain the morphogenesis of the mandibular arch in the developing mouse embryo. They analyze and model the contribution of cell divisions and tissue elasticity in shaping the arch's bud-like protrusion that forms over 4 hours from the 19 to 21 somite stage. Using a combination of live imaging, modeling, and knock-out analysis they propose that cell intercalations are primarily responsible for shaping the arch, and that these cell behaviors are in part coordinated by the influence *Wnt5a*, *Yap/Taz*, and *Piezo1*.

Their theory proposes an intriguing and important (yet often elusive) link between genetic and mechanical forces. However, despite this intrigue the data presented do not strongly support their conclusions, or in some instances directly contradict statements made either due to a lack of statistical analysis or a small sample size. There is a considerable need for improvements before the reader can be convinced of their claims.

1) The use of statistical analysis throughout the paper is haphazard. In some cases, sample size, standard deviations and p-values are calculated and reported, yet in others there is at best a lack of statistical analysis or at worst the reader is told to make a comparison between different conditions in different figures themselves, and in many cases either a conclusion cannot be drawn or what can does not support that of the author's. a. For example, Supplemental Figure 7C claims that *Piezo1* *-/-* cells have lower levels of calcium flux compared to WT, but this is not clear at all from the graphs. The sample size is far too low to make any sense out of the large error bars in Fig 6E or Supp. Fig. 6D for the WT condition, and looks otherwise identical to the *Piezo1* *-/-* mutants. Why do the authors claim that *Wnt5a* *-/-* mutants show a similar level of calcium flux as WT, yet a dampening in *Piezo1* *-/-* mutants? This is a core conclusion in their paper, that is absolutely not convincing from the data that is presented. Furthermore, it is not clear what Movie 29 is supposed to demonstrate. There is some apparent photo-bleaching of the signal, but how is the reader supposed to determine what exactly is being "dampened"? Side-by-side and careful comparisons between WT and mutant cells would help the reader, as well as a larger sample size.

Thank you; we acknowledge the sample size problems outlined by the reviewer and address them individually below.

Regarding calcium fluctuations, we have clarified in the manuscript that the degree of change in calcium reporter intensity is diminished in both *Wnt5a* and *Piezo1* mutants. Additional samples were evaluated and are

shown for WT, *Wnt5a* and *Piezo1* mutants in Supp. Fig. 6D, 7D, E. In Fig. 6 and Supp. Fig. 6, WT and *Wnt5a* mutant graphs are shown side-by-side. However, since there are now ten examples shown for each condition, we have not added the WT graphs alongside the *Piezo1* mutant graphs in Supp. Fig. 7. I take the point about the value of showing WT and mutant data together, but on the other hand some would also take issue with duplication, even in supplementary figures, and Supp. Fig. 7 is especially crowded. In our view, the boxplots representing WT fluctuation are visually (and statistically) distinct from those representing the *Piezo1* mutant. Nonetheless if the reviewer feels strongly, we will add the WT graphs to Supp. Fig. 7D.

Mov. 31 shows that *Piezo1* mutant cells lack calcium fluctuation and as a result the 'stagnant' reporter signal bleaches. It corresponds to Supp. Fig. 7E.

b. Similarly, in Figure 4:

♣ Variation in vinculin force curves between distal and middle mesenchyme is not convincing, a couple of outliers or variation in experiments/embryos in the middle mesenchyme VinTS could be biasing the outcome, and visually is not convincing. This is also true of Figure 6.

Yes, we have added more data for WT (n=5 embryos, 15 cells per embryo per condition) and *Wnt5a* mutant (n=3 embryos, 15 cells per embryo per condition) VinTS comparisons that make the assertion of difference between distal and middle mesenchyme more convincing. Representative curves are updated in Fig. 4E and 6D.

♣ Too few samples for the tailless and donor only plots to comment

Thank you; additional fluorescence lifetime evaluation was performed for both of the control strains (n=3 embryos, 15 cells per embryo per condition), with updated curves shown in Fig. 4E.

♣ Additional questions: was the ratio between VinTS and X-rhod-1 AM staining determined for the entire cell or just for a single plane?

Single planes were used for this evaluation, and this has been disclosed in the methods. We feel this method is justified because the planes were not chosen in any biased fashion and our aim was to compare the temporal variation of calcium reporter intensities, rather than the intracellular spatial distribution of reporter fluorescence.

♣ In Supplementary Fig 4 the authors claim that VinTS lifetime dampened in presence of Y27632 from distal to mid ectoderm, but this does not appear to be not statistically significant at least for the distal ectoderm, it is slightly more convincing for Cyto D, but again the effect appears small: direct comparisons need to be made between WT and treated/mutant embryos, and not ask the reader to compare two separate figures and make a statistical conclusion.

Thank you; contemporaneous data have been combined in a single graph.

c. In general, many effects are small, have too small of a sample size, or are not quantified.

Yes, we have added sample sizes to many experiments quantified all comparisons in the manuscript.

2) There is a lack of scale bars and time bars on many figures and movies. Additionally, many figures and movies are too small to clearly see what is happening.

Thank you; scale bars were added or clarified in Fig. 3E, 4B, 4F, 6A left and right panels, 6E, 7A, 7D, 7H, and Supp. Fig. 1A, 1B, 2A, 2B, 6C.

We enlarged movies 4, 13-15, 17-20, 23, and 26.

Timebars were inserted for movies 5, 12-18, 26-28, 30, 31.

We added the time lapse intervals for light sheet, fluorescence lifetime and cytosolic calcium reporter movies in the legends.

3) The authors use light-sheet imaging to visualize the development of the mandibular arch over a 4 hour period
a. The culture conditions in the Z1 are different than other imaging/incubation conditions, and while the imaging is only done for a relatively short time the impact of this change in conditions on “normal” development is not demonstrated (lower serum, changes in osmolarity, effect of agarose?)

Yes, imaging of intact organ stage mouse embryos by light sheet imaging has not previously been described. We now show how the agarose concentration and presence of serum affect apoptosis and tissue shape in culture over 4 h compared to uncultured embryos in Supp. Fig. 2D, E.

b. Embedding a growing tissue in even 1% agarose has the potential to impose a mechanical constraint on the developing arch, and given the author’s attempts to determine a mechanical component in arch morphogenesis it is important to demonstrate this embedding does not interfere with normal growth. Ideally, if possible, embedding only a portion of the arch in agarose or removing the agarose from just around the arch before placing it in the imaging chamber would eliminate the potential for concern.

Change in the width/length ratio of the arch was slowed by any culture method compared to a similar duration of development *in utero*, and most of all using 2% agarose. We generated an agarose-free region around the face and found that drift was excessive. The width/length outcome of 1% agarose was similar to the tissue being free in medium, so we chose 1% to minimise and facilitate tracking of tissue drift by embedding fluorescent beads alongside the embryo.

c. A z-step size of 0.479 microns is quite a bit of oversampling given a likely light-sheet thickness of 2-4 microns on the Z1 for a sample of that size. A larger step size would help to spare the tissue from excessive illumination.

In our case, the optimal z-spacing was calculated automatically based on the numerical aperture of the objective, which was 1.0, rather than the light-sheet thickness. We have noted this point in the methods. We were fairly pleased with the resolution of nuclei and membranes in 3D for the purpose of this study, but we will certainly take the advice to titrate the z-step size for future analyses.

d. The small field of view in many movies makes it difficult to interpret what is happening. Many of the labels are far too small to read.

Yes; the majority of the movies were edited to add time bars, increase size and adjust frame rate.

e. Without a clear membrane label interpreting intercalation events based on nuclear position alone can be misleading, however given the difficulty in segmenting membranes in crowded environments (particularly with the mT/mG reporter) this is a challenging problem to address. The authors do however briefly show the use of a mosaic myr-Venus reporter that looked fairly nice for visualizing cell membranes/intercalations. More extensive use of this reporter might help to better elucidate/visualize cell behaviors.

Thank you for the suggestion. I’m afraid that use of our mosaic myr-Venus in time lapse movies has not improved our ability to identify intercalations based on membrane rearrangements because cell neighbours are not consistently visible. Of course, the same challenge of rendering membranes in 3D persists with that reporter. Evidence for membrane rearrangements based on the mTmG reporter is shown in movies 5, and 11 and is the main focus of movie 16, although we could provide more like it if the reviewer feels it would be useful.

f. A higher number of tracked nuclei over a larger time-scale would be beneficial to support the author’s conclusions about cell behavior, particularly with regards to intercalation behavior (which is not clear from the small number of nuclei sampled or from the videos). There are a couple registration/drift compensation modules that might be of use to the authors and improve their ability to track cells, particularly since they include beads in their agarose mounting. A free and relatively well-documented example would be the Preibisch multi-view registration plugin for Fiji.

Thank you; we have tracked a larger number of nuclei for WT and *Wnt5a* mutant and have revised Figs. 2 and 5 and the corresponding movies. The rostrocaudal and oblique sweeping movements of mesenchymal cells within the middle region of the arch correspond more convincingly with the small scale intercalations that we show.

4) Additional questions/comments:

a. It could be interesting if one were to block cell divisions in the arch and compare the resulting shape WT to fully elucidate the contribution of cell proliferation to arch shape.

This is a conceptually good suggestion but I'm afraid we cannot think of a pharmacological way to block cell division without affecting other dynamic cell behaviours. A related limitation would be the duration of development that we could observe *ex-utero*. One possibility would be to genetically disrupt the cell cycle more specifically in a temporally conditional manner just as the arch begins to bud, but I'm afraid we can't commit to that experiment at this time.

b. Can the model predict shapes based on cell movements?

This question reflects the current direction of a few different approaches to computational modelling of morphogenesis including one that my lab is developing, although none are ready for application to 3D tissues. The finite element approach we employed treats tissue as a continuum so individual cells cannot be considered. We tried to mimic the effect of cell movements within this continuum model by adding vectors of compression or extension, but the outcome is derivative and coarse and therefore not of any value. The model we are developing will hopefully allow us to address this question properly within the next two years.

c. In several of the mutant OPT images there appears to be at least some narrowing in the waist. If this is due to intercalation as the author's propose, what is the explanation for this narrowing in mutant cases?

There are indeed diminished cell intercalations in *Wnt5a* mutants. The revised cell tracking images shown in Fig. 5 and the corresponding movie support that conclusion, and the 'dandelion' plot suggests that mutant intercalations do not sufficiently drive cells along the long axis of the arch.

Reviewer #4 (Remarks to the Author):

Development of the mammalian mandibular arch during embryogenesis is critical for multiple aspects of lower face development. Abnormalities in this developmental process are associated with a number of human craniofacial birth defects. Although we have a broad understanding of the genetic interactions responsible for shaping the mandibular arch, our understanding of cell behaviour behind these morphogenetic changes in the shape of the arch over time are rudimentary at best. This manuscript uses a variety of approaches to tease apart the cellular properties that lead to the arch having a different shape in its mid region – which is narrower – compared to the distal end, which is more bulbous, during a specific developmental period. The authors find that changes in the rate of cell division, elasticity and viscosity of the tissue alone cannot account for these shape changes. Following on from these observations they determined that there were significant changes in orientated cell movement, cell contacts and cell intercalation behaviors between the mid and distal regions that appear to account for the shape changes. The authors subsequently develop a hypothesis based on cell intercalation behavior and test this idea using a series of mouse mutants. They use elegant molecular genetic approaches to integrate a fluorescence indicator of cortical tension into the mouse embryo and record changes in tension and calcium fluctuations in the different regions of the arch that correlate with the shape differences. Finally, the authors utilize available mouse mutants to link non-canonical Wnt5a signaling through Yap and Piezo to these cell behaviours.

Overall, this is a comprehensive analysis of a specific issue regarding facial morphogenesis that provides an important contribution to our understanding of this process. The study is supported by strong approaches using both genetics and cell biological methodology and the data shown in the main and supplementary figures are for the most part convincing and are supported by extensive movies.

However, there are several problems with the manuscript that warrant attention before it can be considered for publication.

1.

The Discussion is quite brief. If possible the Discussion should place the results found for the mandibular arch in the context of the genetics of human Robinow Syndrome, which can be caused by WNT5A mutations inherited in an autosomal dominant manner. In this respect, the authors should refer to a recent manuscript from Hosseini-Farahabadi et al (JDR, vol 96, pp1265-72 (2017)) that examines aspects of Wnt5a action on mandibular development in the chick. It would also be useful if the data from the current manuscript were placed

into a broader context of mouse mutations that cause similar shape changes in the mandibular arch.

Thank you – we expanded on mutations that cause mandibular anomalies in the introduction and the discussion, focusing on pathways which are addressed genetically in our manuscript. The Hosseini-Farahabadi manuscript has interesting parallels with our work which we highlight in the discussion. We believe this broader context will indeed improve the relevance and appeal of the manuscript.

2.

Similarly, the discussion should include details of how the studies on *Wnt5a* described here for the mandible correlate with and inform other biological processes regulated by *Wnt5a* including limb bud morphogenesis, axial extension, and potentially tumorigenesis to place these studies in a wider context for this gene's role in development, morphogenesis and disease.

Although this request makes sense, I do worry that a fairly broad review of *Wnt5a* in multiple contexts might detract from the biophysical functions that we were very keen to focus on. That is why we have kept the discussion brief, so that the take home points that we regard as important are clear. Also, the manuscript is currently too long. In my view, a separate review of the emerging biophysical role of *Wnt5a* in light of its traditional functions would be beneficial.

3. Results, Methods, Legend and Supp Fig 1. There are three issues regarding cell cycle analysis.

The schematic in Fig Supp 1A does not match the text presented on page 14/15 of the methods either in the order of the injections or the timing between them. This issue needs to be clarified.

Thank you; the schematic and methods have been corrected.

The authors also use both IddU and IdU throughout the manuscript interchangeably. I think the latter is the correct term.

Thank you; Idu has been now used throughout.

The immunological reagent used to detect IdU needs to be listed in the methods section on page 21.

Yes, that antibody has now been included.

4.

Results. The text on page 8 discussing the development of the VinTS, VinTFP, and vinTL would benefit from a short description of how vinculin would provide a readout of tension, particularly with respect to the presence or absence of the tail domain.

Yes, we explained that vinculin links cadherin/catenin complexes at adherens junctions with cortical actin under tension, and that the lack of C-terminal actin binding sites in the tailless knock-in should result in maximal FRET.

5.

The data concerning the calcium signal fluctuations are not as convincing as presented as other studies in the manuscript, and the meaning of “dampening” used in the context of these experiments could be clearer. For example, in Supp Fig 4G, the data are presented as area, but it is not clear if the dampening represents a change in amplitude alone or also reflects that the overall signal intensity is less in the distal mesenchyme.

This point was indeed confusing. We clarified in the results section and in the figure legend that area refers to proportion of X-rhod-1 fluorescence per cell outlined by VinTS in a single confocal plane quantified using Image J.

6.

With respect to the modeling and morphometric measurements in the control and mutants, which employ cell cycle, elasticity and viscosity measurements, the authors should mention whether the volume of the arch remains constant or whether changes are observed in this parameter that could also impact the analysis of underlying cell behaviours.

The finite element model can't account for individual cell behaviours such as intercalation, but does simulate volume change as growth pressure based on our measurements of cell cycle times. This point should now be clear in the results, figure legend and methods.

7.

The methods section does not contain descriptions of the studies in Supp Fig 4 on the embryoid bodies and cardiomyocytes.

Thank you, the derivation of embryoid bodies and cardiomyocytes has now been described and cited in the methods.

8.

Page 10 and Figure 7A.

Further information should be provided concerning the expression of the Sox2:Cre driven Wnt5a construct. The authors say that Wnt5a becomes ubiquitously expressed and disrupts the gradient in the mandible. However, the expression of the endogenous gene seems to be much higher than the transgene from the in situ analysis. It would be helpful if the authors could quantitate how much the relative expression of Wnt5a has changed in the proximal mandibular regions in controls versus transgenics. It would be particularly helpful to know how the increased levels of transgene expression in the embryo as a whole relate to the levels of the endogenous gene expression in the distal mandible (high signal) so that it becomes clearer how the gradient has been affected.

Thank you, we performed quantitative RT-PCR on dissected sections of the mandibular arch from *Sox2:Cre;Z/Wnt5a* embryos and show that data in Supp. Fig. 7A. The principal difference is in the middle arch where WT expression is low.

Minor Points

Page 11.

"In Wnt5a mutants, nuclear localization of YAP", since this sentence comes soon after the discussion of the Wnt5a overexpressing mice, it would be helpful to say "In Wnt5a loss of function mutants" to clarify the strain under discussion.

Thank you, the change has been made.

Page 20.

"A persistent random walk, however, is characterized by slope 1, the short black line in the figure". Please indicate the figure to which this refers.

Yes, Supp. Fig. 2G has now been indicated.

Page 31.

Figure 2 and legend. The meaning of the different colours in panels A, H and I should be included. For example, "Coloured lines correspond to components of the strain tensor" does not provide sufficient information concerning the nature of each line shown. The authors could also refer to a relevant movie in the appropriate place in the legend to strengthen the arguments.

Thank you; for the legend of Fig. 2A it has been stated that the colours represent individual cell neighbours. For Fig. 2H, we added a colour code to identify each of the four rows of square domains of the strain grid along the rostrocaudal axis. Those colours correspond to the coloured, dashed lines in panel I. We added the movie reference in the figure legend as well.

Page 31. Supp Fig 2 and legend.

The meaning of the colours in panel H should be explicitly stated.

Thank you, the colour legend has been added to the figure panel and the legend.

Page 31, Figure 3 and legend.

A time series is mentioned for panel 3, but no time parameters are shown.

Thank you; the duration of those time lapse movies has now been noted in the legend.

Fig. 4, Fig Supp 4 and Fig 6.

I am unclear why the “Lifetime range (max-min) in vivo” is under 1ns in Supp Fig 4D, but the Average lifetime is over 2ns in other figures and panels.

Yes, that was confusing. We changed the presentation of those data in response to another reviewer’s request and have clarified that the range refers to dynamic range of different sensor strains in different spatial regions.

Page 33, Supp Fig 5D Legend.

“In contrast to WT embryos (see Fig. 1F) middle arch epithelial stiffness did not increase significantly”. This should possibly be rephrased to “increase as significantly” since they have asterisks on the figure to show that $P < 0.05$ for these measurements.

Thank you; the suggested change was made.

Page 35, Supp Fig 7C legend.

The meaning of the individual panels #1, #2 and #3 at the bottom of the figure is not given.

Thank you; it has been written now that each graph represents one cell.

Page 37, Movie 25.

The legend refers to Fig 5E and Supp Fig 5D, but these are not the correct figures.

Thank you; the correction has been made.

Typos:

Page 13: “Animal care Committ,ee”

Page 15: “The Possion’s ratio”

Page 31: “According to a the model” and “tranjectories”

Supp Fig 2, panel G “Tima lag”

Page 33, Supp Fig 5 Legend. “Between 19 and 20 somite stages” Should be between 19 and 21.

Thank you, the appropriate changes were made. We very much appreciate the reviewer’s attention to detail.

Reviewers' Comments:

Reviewer #1:

Remarks to the Author:

This is a revised version of an earlier submission describing the changes in the mouse mandibular tissue from stage 19-21. They follow the cell movements with nuclear markers in planar sheet microscopy and show that contraction in the middle portion of the tissue causes elongation. As predicted, a vinculin tension sensor (using FRET) shows greater forces in the middle region that correlate with increased calcium entry there. Interestingly, they find oscillations in the level of FRET on the time scale of about 5 minutes indicating that the motility is driven by cyclic contractions. Mutants in Wnt5a, Yap and Piezo1 proteins result in shortened tissues. They propose a pathway for the morphological changes with these players.

This work will stimulate experiments on the detailed motions of the cells during a simple developmental process and they have identified several players that can be followed. In the revision of the manuscript, they seem to have addressed the concerns of the previous reviewers and I recommend acceptance.

Reviewer #3:

Remarks to the Author:

The authors have made a substantial effort to address claims and concerns by the reviewers, including increasing sample size and providing more statistical comparisons. The data are much more convincing and easier for the reader to interpret as a result (although I'm still not entirely convinced by 6E), and largely support their conclusions. There is certainly a large amount of follow-up work that could be performed, however for its current format is already quite sufficient and publication of this manuscript will hopefully result in some useful discussions in the field.

Reviewer #4:

Remarks to the Author:

The revised manuscript by Tao et al has addressed all my major comments from the previous review. It is an interesting article that deserves publication.

Minor textual issue:

Figure 5 legend, D, E. The authors say "persistence of cell movements (persistent time, (B)) and direction (angle from mean, (C))". I think this should be (D) and (E) instead of (B) and (C).